theoretical biology, evolution, computational biology

reproductive division of labour, origin of genetic information, RNA world hypothesis, prebiotic evolution, Price equation

**Author for correspondence:**
Nobuto Takeuchi
e-mail: nobuto.takeuchi@auckland.ac.nz

# The origin of the central dogma through conflicting multilevel selection

Nobuto Takeuchi[1,3] and Kunihiko Kaneko[1,2]

[1]Research Center for Complex Systems Biology, and [2]Department of Basic Science, Graduate School of Arts and Sciences, University of Tokyo, Komaba 3-8-1, Meguro-ku, Tokyo 153-8902, Japan
[3]School of Biological Sciences, Faculty of Science, University of Auckland, Private Bag 92019, 1142 Auckland, New Zealand

(iD) NT, 0000-0003-4949-6476

The central dogma of molecular biology rests on two kinds of asymmetry between genomes and enzymes: informatic asymmetry, where information flows from genomes to enzymes but not from enzymes to genomes; and catalytic asymmetry, where enzymes provide chemical catalysis but genomes do not. How did these asymmetries originate? Here, we show that these asymmetries can spontaneously arise from conflict between selection at the molecular level and selection at the cellular level. We developed a model consisting of a population of protocells, each containing a population of replicating catalytic molecules. The molecules are assumed to face a trade-off between serving as catalysts and serving as templates. This trade-off causes conflicting multilevel selection: serving as catalysts is favoured by selection between protocells, whereas serving as templates is favoured by selection between molecules within protocells. This conflict induces informatic and catalytic symmetry breaking, whereby the molecules differentiate into genomes and enzymes, establishing the central dogma. We show mathematically that the symmetry breaking is caused by a positive feedback between Fisher's reproductive values and the relative impact of selection at different levels. This feedback induces a division of labour between genomes and enzymes, provided variation at the molecular level is sufficiently large relative to variation at the cellular level, a condition that is expected to hinder the evolution of altruism. Taken together, our results suggest that the central dogma is a logical consequence of conflicting multilevel selection.

## 1. Introduction

At the heart of living systems lies a distinction between genomes and enzymes—a division of labour between the transmission of genetic information and the provision of chemical catalysis. This distinction rests on two types of asymmetry between genomes and enzymes: informatic asymmetry, where information flows from genomes to enzymes but not from enzymes to genomes; and catalytic asymmetry, where enzymes provide chemical catalysis but genomes do not. These two asymmetries constitute the essence of the central dogma in functional terms [1].

However, current hypotheses about the origin of life posit that genomes and enzymes were initially undistinguished, both embodied in a single type of molecule, RNA or its analogues [2]. While these hypotheses resolve the chicken-and-egg paradox of whether genomes or enzymes came first, they raise an obvious question: how did the genome-enzyme distinction originate?

Michod hypothesized that a genome-enzyme distinction evolved because the distinction maximized the multiplication rates of replicators by allowing the unconstrained optimization of the replication rate and hydrolytic resistance of replicators that are in a trade-off relation [3].

We consider an alternative possibility that does not depend on the assumption that a genome-enzyme distinction maximized the multiplication rates of

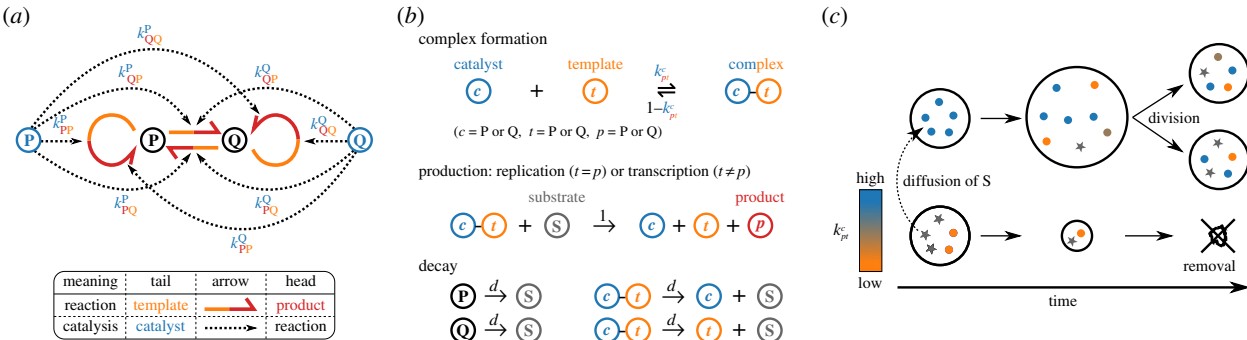

**Figure 1.** The agent-based model (see Methods for details). (*a*) Two types of replicators, P and Q, can serve as templates and catalysts for producing either type. Circular harpoons indicate replication; straight harpoons, transcription (heads indicate products; tails, templates). Dotted arrows indicate catalysis (heads indicate reaction catalysed; tails, replicators providing catalysis). (*b*) Replicators undergo complex formation, replication, transcription, and decay. Rate constants of complex formation are given by the $k_{pt}^c$ values of a replicator serving as a catalyst (whose type, P or Q, is denoted by $c$). The catalyst can form two distinct complexes with another replicator serving as a template (whose type is denoted by $t$) depending on whether it replicates ($p = t$) or transcribes ($p \neq t$) the template. (*c*) Protocells exchange substrate (represented by stars) through rapid diffusion. Protocells divide when the number of internal particles exceeds *V*. Protocells are removed when they lose all particles. (Online version in colour.)

replicators. Specifically, we explore the possibility that a genome-enzyme distinction arose from conflict between selection at the level of protocells and selection at the level of molecules within protocells. During the evolutionary transition from replicating molecules to protocells, competition occurred both between protocells and between molecules within protocells [4–7]. Consequently, selection operated at both cellular and molecular levels, and selection at one level was potentially in conflict with selection at the other [8,9]. Previous studies have demonstrated that such conflicting multilevel selection can induce a partial and primitive distinction between genomes and enzymes in replicating molecules [10,11]. Specifically, the molecules undergo catalytic symmetry breaking between their complementary strands, whereby one strand becomes catalytic and the other becomes non-catalytic. However, the molecules do not undergo informatic symmetry breaking—i.e. one-way flow of information from non-catalytic to catalytic molecules—because complementary replication necessitates both strands to be replicated. Therefore, the previous studies have left the most essential aspect of the central dogma unexplained.

Here, we investigate whether conflicting multilevel selection can induce both informatic and catalytic symmetry breaking in replicating molecules. To this end, we extend the previous model by considering two types of replicating molecules, denoted by P and Q. Although P and Q could be interpreted as RNA and DNA, their chemical identity is unspecified for simplicity and generality. To examine the possibility of spontaneous symmetry breaking, we assume that P and Q initially do not distinguish each other. We then ask whether evolution creates a distinction between P and Q such that information flows irreversibly from one type (either P or Q) that is non-catalytic to the other that is catalytic.

## 2. Model

Our model is an agent-based model with two types of replicators, P and Q. We assume that both P and Q are initially capable of catalysing four reactions at an equal rate: the replication of P, replication of Q, transcription of P to Q, and transcription of Q to P, where complementarity is ignored (figure 1*a*; note that this

figure does not depict a two-member hypercycle because in our model replicators undergo transcription [12]; see Discussion for more on comparison with hypercycles).

Replicators compete for a finite supply of substrate denoted by S (hereafter, P, Q, and S are collectively called particles). S is consumed through the replication and transcription of P and Q, and recycled through the decay of P and Q (figure 1*b*). Thus, the total number of particles, i.e. the sum of the total numbers of P, Q, and S is kept constant (the relative frequencies of P, Q, and S are variable).

All particles are compartmentalized into protocells, across which P and Q do not diffuse at all, but S diffuses rapidly (figure 1*c*; Methods). This difference in diffusion induces the passive transport of S from protocells in which S is converted into P and Q slowly, to protocells in which this conversion is rapid. Consequently, the latter grow at the expense of the former [13]. If the number of particles in a protocell exceeds threshold *V*, the protocell is divided with its particles randomly distributed between the two daughter cells; conversely, if this number decreases to zero, the protocell is discarded.

Crucial in our modelling is the incorporation of a trade-off between a replicator's catalytic activities and templating opportunities. This trade-off arises from the constraint that providing catalysis and serving as a template impose structurally incompatible requirements on replicators [14,15]. Because replication or transcription takes a finite amount of time, serving as a catalyst comes at the cost of spending less time serving as a template, thereby inhibiting replication of itself. To incorporate this trade-off, the model assumes that replication and transcription entail complex formation between a catalyst and template (figure 1*b*) [16]. The rate constants of complex formation are given by the catalytic activities (denoted by $k_{pt}^c$) of replicators, as described below.

Each replicator is individually assigned eight catalytic values denoted by $k_{pt}^c \in [0, 1]$, where the indices ($c$, $p$, and $t$) are P or Q (figure 1*a*). Four of these $k_{pt}^c$ values denote the catalytic activities of the replicator itself; the other four, those of its transcripts. For example, if a replicator is of type P, its catalytic activities are given by its $k_{pt}^P$ values, whereas those of its transcripts, which are of type Q, are given by its $k_{pt}^Q$ values. The indices $p$ and $t$ denote the specific type of reaction catalysed, as depicted in figure 1*a*. When a new replicator is

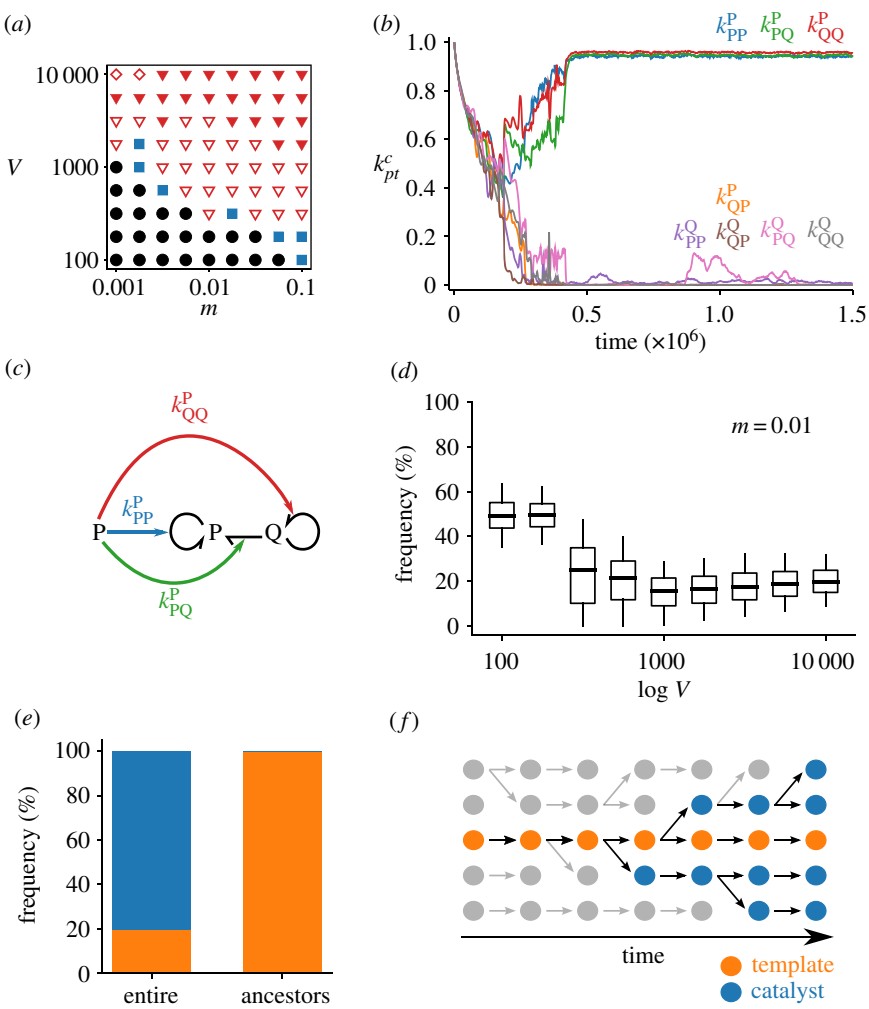

**Figure 2.** The evolution of the central dogma. (*a*) Phase diagram: circles indicate no symmetry breaking (electronic supplementary material, figure S1*a,b*); squares, uncategorized (electronic supplementary material, figure S1*c,d*); open triangles, incomplete symmetry breaking (electronic supplementary material, figure S1*e–h*); filled triangles, threefold symmetry breaking as depicted in *b, c,* and *d*; diamonds, catalytic and informatic symmetry breaking without numerical symmetry breaking (electronic supplementary material, figure S5*a*). The initial condition was $k_{pt}^c = 1$ for all replicators. (*b*) Dynamics of $k_{pt}^c$ averaged over all replicators. $V = 10\,000$ and $m = 0.01$. (*c*) Catalytic activities evolved in b. (*d*) Per-cell frequency of minority replicator types (P or Q) at equilibrium as a function of $V$: boxes, quartiles; whiskers, 5th and 95th percentiles. Only protocells containing at least $V/2$ particles were considered. (*e*) Frequencies of templates (orange) and catalysts (blue) in the entire population or in the common ancestors. $V = 3162$ and $m = 0.01$. (*f*) Illustration of e. Circles represent replicators; arrows, genealogy. Extinct lineages are grey. Common ancestors are always templates, whereas the majority of replicators are catalysts. (Online version in colour.)

produced, its $k_{pt}^c$ values are inherited from its template with potential mutation of probability $m$ (Methods).

The $k_{pt}^c$ values of a replicator determine the rates at which this replicator forms a complex with another replicator and catalyses replication or transcription of the latter (figure 1*b*; Methods). The greater the catalytic activities ($k_{pt}^c$) of a replicator, the greater the chance that the replicator is sequestered in a complex as a catalyst and thus unable to serve as a template—hence a trade-off. Note that the trade-off is relative: if all replicators in a protocell have identical $k_{pt}^c$ values, their multiplication rate increases monotonically with their $k_{pt}^c$ values, assuming all else is held constant.

The above trade-off creates a dilemma: providing catalysis brings benefit at the cellular level because it accelerates a protocell's uptake of substrate; however, providing catalysis brings cost at the molecular level because it decreases the relative opportunity of a replicator to be replicated within a protocell [10]. Therefore, selection between protocells tends to maximize the $k_{pt}^c$ values of replicators (i.e. cellular-level selection), whereas selection within protocells tends to minimize the $k_{pt}^c$ values of replicators (i.e. molecular-level selection).

## 3. Results

### (a) Computational analysis

Using the agent-based model described above, we examined how $k_{pt}^c$ values evolve as a result of conflicting multilevel selection. To this end, we set the initial $k_{pt}^c$ values of all replicators to 1, so that P and Q are initially identical in their catalytic activities (the initial frequencies of P or Q are also set to be equal). We then simulated the model for various values of $V$ (the threshold at which protocells divide) and $m$ (mutation rate).

Our main result is that for sufficiently large values of $V$ and $m$, replicators undergo spontaneous symmetry breaking in three aspects (figure 2*a–d*; electronic supplementary material, figure S1). First, one type of replicator (either P or Q) evolves high catalytic activity, whereas the other completely loses it (i.e. $k_{pt}^c \gg k_{pt}^{c'} \approx 0$ for $c \neq c'$): catalytic symmetry breaking (figure 2*b,c*). Second, templates are transcribed into catalysts, but catalysts are not reverse-transcribed into templates (i.e. $k_{ct}^c \gg k_{tc}^c \approx 0$): informatic symmetry breaking (figure 2*b,c*). Finally, the copy number of templates becomes

smaller than that of catalysts: numerical symmetry breaking (figure 2d). This threefold symmetry breaking is robust to various changes in model details (see electronic supplementary material, Text 1.1 and 1.2; figures S2–S4).

A significant consequence of the catalytic and informatic symmetry breaking is the resolution of the dilemma between providing catalysis and getting replicated. Once symmetry is broken, tracking lineages reveals that the common ancestors of all replicators are almost always templates (figure 2e,f; Methods). That is, information is transmitted almost exclusively through templates, whereas information in catalysts is eventually lost (i.e. catalysts have zero reproductive value). Consequently, evolution operates almost exclusively through competition between templates, rather than between catalysts. How the catalytic activity of catalysts evolves, therefore, depends solely on the cost and benefit to templates. On one hand, this catalytic activity brings benefit to templates for competition across protocells. On the other hand, this activity brings no cost to templates for competition within a protocell (neither does it bring benefit because catalysis is equally shared among templates). Therefore, the catalytic activity of catalysts is maximized by cellular-level selection operating on templates, but not minimized by molecular-level selection operating on templates, hence the resolution of the dilemma between catalysing and templating. Because of this resolution, symmetry breaking leads to the maintenance of high catalytic activities (electronic supplementary material, figures S6 and S7).

## (b) Mathematical analysis

To understand the mechanism of the catalytic and informatic symmetry breaking, we simplified the agent-based model into mathematical equations. These equations allow us to consider all the costs and benefits involved in the provision of catalysis by $c \in \{P, Q\}$: molecular-level cost to $c$ (denoted by $\gamma_c^c$) and cellular-level benefit to $t \in \{P, Q\}$ (denoted by $\beta_c^t$). The equations calculate the joint effects of all these costs and benefits on the evolution of the average catalytic activities of $c$ (denoted by $\bar{k}^c$). The equations are derived with the help of Price's theorem [8,9,17] and displayed below (see Methods and electronic supplementary material, Text 1.3 for the derivation):

$$
\left.
\begin{array}{l}
\Delta \bar{k}^P \approx \bar{\omega}^P \left( \beta_P^P \sigma_{cel}^2 - \gamma_P^P \sigma_{mol}^2 \right) + \bar{\omega}^Q \beta_P^Q \sigma_{cel}^2 \\
\text{and} \quad \Delta \bar{k}^Q \approx \bar{\omega}^P \beta_Q^P \sigma_{cel}^2 + \bar{\omega}^Q \left( \beta_Q^Q \sigma_{cel}^2 - \gamma_Q^Q \sigma_{mol}^2 \right),
\end{array}
\right\}
\tag{3.1}
$$

where $\Delta$ denotes evolutionary change per generation, $\bar{\omega}^c$ is the average normalized reproductive value of $c$, $\sigma_{cel}^2$ is the variance of catalytic activities among protocells (cellular-level variance), and $\sigma_{mol}^2$ is the variance of catalytic activities within a protocell (molecular-level variance).

The first and second terms on the right-hand side of equations (3.1) represent evolution arising through the replication of P and Q, respectively, weighted by the reproductive values, $\bar{\omega}^P$ and $\bar{\omega}^Q$. The terms multiplied by $\beta_c^t \sigma_{cel}^2$ represent evolution driven by cellular-level selection; those by $-\gamma_c^c \sigma_{mol}^2$, evolution driven by molecular-level selection.

The derivation of equations (3.1) involves various simplifications that are not made in the agent-based model, among which the three most important are noted below (see Methods and electronic supplementary material, Text 1.3 for details). First, equations (3.1) simplify evolutionary dynamics by restricting the number of evolvable parameters to a minimum required for catalytic and informatic symmetry breaking. More specifically, equations (3.1) assume that $k_{pt}^c$ is independent of $p$ and $t$ (denoted by $k^c$), i.e. catalysts do not distinguish the replicator types of templates and products. Despite this simplification, catalytic symmetry breaking can still occur (e.g. $k^P > k^Q$), as can informatic symmetry breaking: the trade-off between catalysing and templating causes information to flow preferentially from less catalytic to more catalytic replicator types. However, numerical symmetry breaking is excluded as it requires $k_{pt}^c$ to depend on $p$; consequently, the frequencies of P or Q are fixed and even in equations (3.1) (this is not the case in the agent-based model described in the previous section). Therefore, while equations (3.1) are useful for identifying the mechanism of catalytic and informatic symmetry breaking, they are not useful for identifying the mechanism of numerical symmetry breaking. In the electronic supplementary material, we use different equations to identify the mechanism of numerical symmetry breaking (see electronic supplementary material, Text 1.4 and figure S5).

The second simplification involved in equations (3.1) is that variances $\sigma_{mol}^2$ and $\sigma_{cel}^2$ are treated as parameters although they are actually dynamic variables dependent on $m$ and $V$ in the agent-based model (in electronic supplementary material, we examine this assumption; see electronic supplementary material, Text 1.5 and figure S8). In addition, these variances are assumed to be identical between $\bar{k}^P$ and $\bar{k}^Q$ because no difference is *a priori* assumed between P and Q.

The third simplification involved in equations (3.1) is that the terms of order greater than $\sigma_{cel}^2$ and $\sigma_{mol}^2$ are ignored under the assumption of weak selection [17].

Using equations (3.1), we can now elucidate the mechanism of the symmetry breaking. Consider a symmetric situation where P and Q are equally catalytic: $\bar{k}^P = \bar{k}^Q$. Since P and Q are identical, the catalytic activities of P and Q evolve identically: $\Delta \bar{k}^P = \Delta \bar{k}^Q$. Next, suppose that P becomes slightly more catalytic than Q for whatever reason, e.g. by genetic drift: $\bar{k}^P > \bar{k}^Q$ (catalytic asymmetry). The trade-off between catalysing and templating then causes P to be replicated less frequently than Q, so that $\bar{\omega}^P < \bar{\omega}^Q$ (informatic asymmetry). Consequently, the second terms of equations (3.1) increase relative to the first terms. That is, for catalysis provided by P (i.e. $\bar{k}^P$), the impact of cellular-level selection through Q (i.e. $\bar{\omega}^Q \beta_P^Q \sigma_{cel}^2$) increases relative to those of molecular-level and cellular-level selection through P (i.e. $-\bar{\omega}^P \gamma_P^P \sigma_{mol}^2$ and $\bar{\omega}^P \beta_P^P \sigma_{cel}^2$, respectively), resulting in the relative strengthening of cellular-level selection. By contrast, for catalysis provided by Q (i.e. $\bar{k}^Q$), the impacts of molecular-level and cellular-level selection through Q (i.e. $-\bar{\omega}^Q \gamma_Q^Q \sigma_{mol}^2$ and $\bar{\omega}^Q \beta_Q^Q \sigma_{cel}^2$, respectively) increase relative to that of cellular-level selection through P (i.e. $\bar{\omega}^P \beta_Q^P \sigma_{cel}^2$), resulting in the relative strengthening of molecular-level selection. Consequently, a small difference between $\bar{k}^P$ and $\bar{k}^Q$ leads to $\Delta \bar{k}^P > \Delta \bar{k}^Q$, the amplification of the initial difference—hence, symmetry breaking. The above mechanism can be summarized as a positive feedback between reproductive values and the relative impact of selection at different levels.

We next asked whether, and under what conditions, the above feedback leads to symmetry breaking such that either P or Q completely loses catalytic activity. To address this question, we performed a phase-plane analysis of equations (3.1) as described in figure 3 (see Methods and electronic supplementary material, Text 1.6 for details). Figure 3 shows that

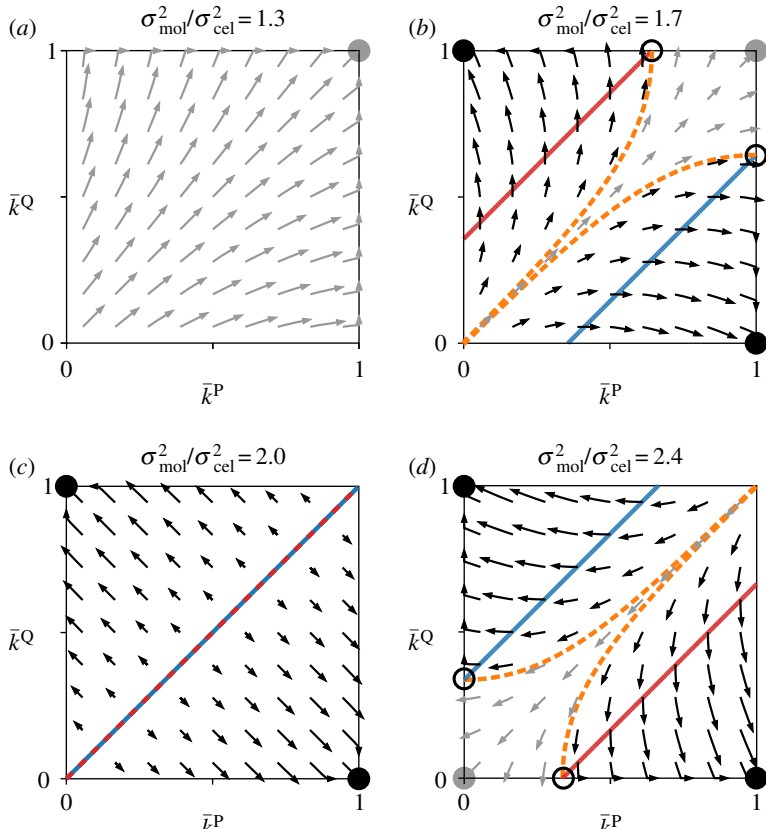

**Figure 3.** Phase-plane analysis. For this analysis, equations (3.1) were adapted as follows: $\beta_c^t$ and $\gamma_c^t$ were set to 1; $\bar{\omega}^c$ was calculated as $e^{-\bar{k}_c}/(e^{-\bar{k}_P} + e^{-\bar{k}_Q})$; $\Delta$ was replaced with time derivative ($d/d\tau$); and $(d/d\tau)\bar{k}^c$ was set to 0 if $\bar{k}^c = 0$ or $\bar{k}^c = 1$ to ensure that $\bar{k}^c$ is bounded within [0, 1] as in the agent-based model. Solid lines indicate nullclines: $(d/d\tau)\bar{k}^P = 0$ (red) and $(d/d\tau)\bar{k}^Q = 0$ (blue). The nullclines at $\bar{k}^c = 0$ and $\bar{k}^c = 1$ are not depicted for visibility. Filled circles indicate symmetric (grey) and asymmetric (black) stable equilibria; open circles, unstable equilibria; arrows, short-duration flows ($\Delta\tau = 0.15$) leading to symmetric (grey) or asymmetric (black) equilibria. Dashed lines (orange) demarcate basins of attraction. $\sigma_{cel}^2 = 1$. (a) Molecular-level variance is so small that cellular-level selection completely dominates; consequently, $\bar{k}^c$ is always maximized. (b) Molecular-level variance is large enough to create asymmetric equilibria; however, cellular-level variance is still large enough to make $\bar{k}^P = \bar{k}^Q = 1$ stable. (c) A tipping point; the nullclines overlap. (d) Molecular-level variance is so large that $\bar{k}^P = \bar{k}^Q = 1$ is unstable; the asymmetric equilibria can be reached if $\bar{k}^P \approx \bar{k}^Q \approx 1$. (Online version in colour.)

$\bar{k}^P$ and $\bar{k}^Q$ diverge from symmetric states (i.e. $\Delta\bar{k}^P \neq \Delta\bar{k}^Q$), confirming the positive feedback described above. However, symmetry breaking occurs only if molecular-level variance $\sigma_{mol}^2$ is sufficiently large relative to cellular-level variance $\sigma_{cel}^2$ (i.e. if genetic relatedness between replicators, $\sigma_{cel}^2/(\sigma_{mol}^2 + \sigma_{cel}^2)$, is sufficiently low; see Methods). Large $\sigma_{mol}^2/\sigma_{cel}^2$ is required because if $\sigma_{mol}^2/\sigma_{cel}^2$ is too small, cellular-level selection completely dominates over molecular-level selection, maximizing both $\bar{k}^P$ and $\bar{k}^Q$ (figure 3a). The requirement of large $\sigma_{mol}^2/\sigma_{cel}^2$ is consistent with the fact that the agent-based model displays symmetry breaking for sufficiently large $V$: the law of large numbers implies that $\sigma_{mol}^2/\sigma_{cel}^2$ increases with $V$ [10,18]. This consistency with the agent-model suggests that equations (3.1) correctly describe the mechanism of symmetry breaking in the agent-based model (see electronic supplementary material, Text 1.5 and figure S8 for an additional consistency check in terms of both $m$ and $V$).

## 4. Discussion

Our results show that conflicting multilevel selection can induce informatic and catalytic symmetry breaking in replicating molecules. The symmetry breaking is induced because molecular-level selection minimizes the catalytic activity of

one type of molecule (either P or Q), whereas cellular-level selection maximizes that of the other. The significance of the symmetry breaking is that it results in the one-way flow of information from non-catalytic to catalytic molecules—the central dogma. The symmetry breaking thereby establishes a division of labour between the transmission of genetic information and the provision of chemical catalysis. This division of labour resolves a dilemma between templating and catalysing, the very source of conflict between levels of selection. Below, we discuss our results in relation to four subjects, namely, chemistry, hypercycle theory, kin selection theory, and Michod's 1983 paper [3].

Our theory does not specify the chemical details of replicating molecules, and this abstraction carries two implications. First, our theory suggests that the central dogma, if formulated in functional terms, is a general feature of living systems that is independent of protein chemistry. When the central dogma was originally proposed, it was formulated in chemical terms as the irreversible flow of information from nucleic acids to proteins [1]. Accordingly, the chemical properties of proteins have been considered integral to the central dogma [19]. By contrast, the present study formulates the central dogma in functional terms, as the irreversible flow of information from non-catalytic to catalytic molecules. Our theory shows that the central dogma, formulated as such, is a logical consequence of conflicting multilevel selection. Therefore, the central dogma might be

a general feature of life that is independent of the chemical specifics of material in which life is embodied.

The second implication of the chemical abstraction is that our theory could be tested by experiments with existing materials. Our theory assumes that a replicator faces a trade-off between providing 'catalysis' and getting replicated. However, it does not restrict catalysis to being replicase activity: although our agent-based model assumes that catalysts are replicases, our mathematical analysis does not. Therefore, existing RNA and DNA molecules could be used to test our theory [20]. For example, one could compare two systems, one where RNA serves as both templates and catalysts, and one where RNA serves as catalysts and DNA serves as templates. According to our theory, the latter is expected to maintain higher catalytic activity through evolution, provided the mutation rate and the number of molecules per cell are sufficiently large (see also [21]). In addition, using RNA and DNA is potentially relevant to the historical origin of the central dogma, given the possibility that DNA might have emerged before the advent of protein translation [22–25].

While our theory is similar to hypercycle theory in that both are concerned with the evolution of complexity in replicator systems, our theory proposes a distinct mechanism for evolving such complexity. Whereas hypercycle theory proposes symbiosis between multiple lineages of replicators [12], our theory proposes symmetry breaking (i.e. differentiation) in a single lineage of replicators—a fundamental distinction that is drawn between 'egalitarian' and 'fraternal' major evolutionary transitions as defined by Queller [26] (egalitarianism implies equality, which is involved in the evolution of complexity through symbiosis, whereas fraternalism implies kinship, which is involved in the evolution of complexity through differentiation; these terms are taken from a French Revolutionary slogan, *Liberté, Egalité, Fraternité*).

Moreover, our theory differs from hypercycle theory in terms of the roles played by non-catalytic templates. In hypercycle theory, the evolution of non-catalytic templates jeopardizes hypercycles because such templates (called parasites) can replicate faster than catalytic templates constituting the hypercycles [16,27]. In our theory, the evolution of non-catalytic templates is one of the essential factors leading to the division of labour between genomes and enzymes.

While our theory differs from hypercycle theory in the above aspects, it does not contradict the latter. In fact, there is a potential synergy between the evolution of complexity through symmetry breaking and that through symbiosis. Our theory posits that a distinction between genomes and enzymes resolves the dilemma between templating and catalysing, thereby increasing the evolutionary stability of catalytic activities in replicators. Likewise, this distinction might also contribute to the evolutionary stability of symbiosis between replicators, hence the potential synergy (however, we should add that the specific mechanism of symbiosis proposed by hypercycle theory is not unique [28–34]).

While our theory is consistent with kin selection theory, it makes a novel prediction for evolution under a condition of low genetic relatedness. Kin selection theory posits that altruism can evolve if genetic relatedness is sufficiently high [35]. Consistent with this, our theory posits that for sufficiently high genetic relatedness (i.e. for sufficiently high $\sigma_{cell}^2/(\sigma_{cel}^2 + \sigma_{mol}^2)$, or sufficiently small $m$ and $V$), cellular-level selection maximizes the

provision of catalysis by all molecules, establishing full altruism (providing catalysis can be viewed as altruism [36]: providing catalysis brings no direct benefit to a catalyst because a catalyst cannot catalyse the replication of itself in our model). However, the two theories diverge for sufficiently low genetic relatedness. In this case, kin selection theory predicts that evolution cannot lead to altruism; by contrast, our theory predicts that evolution can lead to a division of labour between the transmission of genetic information and the provision of chemical catalysis. Whether this reproductive division of labour should be called altruism is up for debate.

Michod hypothesized that a genome-enzyme distinction evolved because the distinction maximized the multiplication rates of replicators by allowing the unconstrained optimization of the replication rate and hydrolytic resistance of replicators that are in a trade-off relation [3]. While our present work is similar to Michod's in underlining trade-off faced by replicators, it differs from the latter in two aspects. First, our work provides a model that explicitly demonstrates the evolution of a genome-enzyme distinction, whereas Michod's work does not (the latter instead describes mathematical modelling that examines a condition required for the invasion of a hypercycle; however, the invasion of a hypercycle does not necessarily imply the evolution of a genome-enzyme distinction).

Second, Michod's hypothesis assumes that a genome-enzyme distinction maximizes the multiplication rates of replicators, whereas our model does not involve this assumption. In our model, the multiplication rates of replicators increase monotonically with their catalytic activities if replicators have identical catalytic activities, assuming all else is held constant. Therefore, the multiplication rates of replicators are maximized if all replicators are maximally catalytic, a state that involves no genome-enzyme distinction. This state, in fact, evolves for sufficiently small $V$ and $m$ values, i.e. for sufficiently high relatedness (see also the discussion of kin selection theory above).

One might wonder how our model could display the evolution of a genome-enzyme distinction without the assumption that a genome-enzyme distinction maximizes the multiplication rates of replicators. The answer is conflicting multilevel selection. In our model, a genome-enzyme distinction evolves because it is a stable equilibrium between evolution driven by molecular-level selection and evolution driven by cellular-level selection. The symmetry breaking that creates this distinction cannot be induced by selection at a single level, molecular or cellular, because selection at a single level either maximizes or minimizes all catalytic activities of all replicators. Rather, the symmetry breaking is induced by conflict between molecular-level selection and cellular-level selection, the interaction that creates a positive feedback between reproductive values and the relative impact of selection at different levels. Similar results have been obtained from previous studies, where interactions between conflicting levels of selection are shown to evolve various states that are not directly selected for at any single level [10,21,37]. Taken together, these results suggest the possibility that biological complexity evolves as emergent outcomes of conflicting multilevel selection.

Finally, we note that the division of labour between the transmission of genetic information and other functions is a recurrent pattern throughout biological hierarchy. For example, multicellular organisms display differentiation

**Table 1.** Division of labour between information transmission and other functions transcends the levels of biological hierarchy.

| hierarchy | | differentiation | |
| --- | --- | --- | --- |
| **whole** | **parts** | **information** | **other** |
| cell | molecules | genome | enzyme |
| symbiont population* | prokaryotic cells | transmitted | non-transmitted |
| ciliate | organelles | micronucleus | macronucleus |
| multicellular organism | eukaryotic cells | germline | soma |
| eusocial colony | animals | queen | worker |

*Bacterial endosymbionts of ungulate lice (*Haematopinus*) and planthoppers (*Fulgoroidea*) [38].

between germline and soma, as do eusocial animal colonies between queens and workers (table 1) [4–7]. Given that all these systems potentially involve conflicting multilevel selection and tend to display reproductive division of labour as their sizes increase [7], our theory might provide a basis on which to pursue a universal principle of life that transcends the levels of biological hierarchy.

## 5. Methods

### (a) Details of the model

The model treats each molecule as a distinct individual with uniquely assigned $k_{pt}^c$ values. One time step of the model consists of three substeps: reaction, diffusion, and cell division.

In the reaction step, the reactions depicted in figure 1b are simulated with the algorithm described previously [10]. The rate constants of complex formation are given by the $k_{pt}^c$ values of a replicator serving as a catalyst. For example, if two replicators, denoted by $X$ and $Y$, serve as a catalyst and template, respectively, the rate constant of complex formation is the $k_{py}^x$ value of $X$, where $x$, $y$, and $p$ are the replicator types (i.e. P or Q) of $X$, $Y$, and product, respectively. If $X$ and $Y$ switch the roles (i.e. $X$ serves as a template, and $Y$ serves as a catalyst), the rate constant of complex formation is the $k_{px}^y$ value of $Y$. Complexes are distinguished not only by the roles of $X$ and $Y$, but also by the replicator type of product $p$. Therefore, $X$ and $Y$ can form four distinct complexes depending on which replicator serves as a catalyst and which type of replicator is being produced.

The above rule about complex formation implies that whether a template is replicated ($p = t$) or transcribed ($p \neq t$) depends entirely on the $k_{pt}^c$ values of a catalyst. In other words, a template cannot control how its information is used by a catalyst. This rule excludes the possibility that a template maximizes its fitness by biasing catalysts towards replication rather than transcription. Excluding this possibility is legitimate if the backbone of a template does not directly determine the backbone of a product as in nucleic acid polymerization.

In addition, the above rule about complex formation implies that replicators multiply fastest if their $k_{pt}^c$ values are maximized for all combinations of $c$, $p$, and $t$ (this is because $X$ and $Y$ form a complex at a rate proportional to $\sum_p k_{py}^x + k_{px}^y$ if all possible complexes are considered). Consequently, cellular-level selection tends to maximize $k_{pt}^c$ values for all combinations of $c$, $p$, and $t$ (because cellular-level selection tends to maximize the multiplication rate of replicators within protocells). If $k_{pt}^c$ values are maximized for all combinations of $c$, $p$, and $t$, P and Q coexist. Therefore, coexistence between P and Q is favoured by cellular-level selection, a situation that might not always be the case in reality. We ascertained that the above specific rule about complex

formation does not critically affect results by examining an alternative model in which cellular-level selection does not necessarily favour coexistence between P and Q (see electronic supplementary material, Text 1.1).

In the diffusion step, all substrate molecules are randomly redistributed among protocells with probabilities proportional to the number of replicators in protocells. In other words, the model assumes that substrate diffuses extremely rapidly.

In the cell-division step, every protocell containing more than $V$ particles (i.e. P, Q, and S together) is divided as described in Model.

The mutation of $k_{pt}^c$ is modelled as unbiased random walks. With a probability $m$ per replication or transcription, each $k_{pt}^c$ value of a replicator is mutated by adding a number randomly drawn from a uniform distribution on the interval $(-\delta_{mut}, \delta_{mut})$ ($\delta_{mut} = 0.05$ unless otherwise stated). The values of $k_{pt}^c$ are bounded above by $k_{max}$ with a reflecting boundary ($k_{max} = 1$ unless otherwise stated), but are not bounded below to remove the boundary effect at $k_{pt}^c = 0$. However, if $k_{pt}^c < 0$, the respective rate constant of complex formation is regarded as zero.

We ascertained that the above specific model of mutation does not critically affect results by testing two alternative models of mutation. One model is nearly the same as the above, except that the boundary condition at $k_{pt}^c = 0$ was set to reflecting. The other model implements mutation as unbiased random walks on a logarithmic scale. The details are described in electronic supplementary material, Text 1.2.

Each simulation was run for at least $5 \times 10^7$ time steps (denoted by $t_{min}$) unless otherwise stated, where the unit of time is defined as that in which one replicator decays with probability $d$ (thus, the average lifetime of replicators is $1/d$ time steps). The value of $d$ was set to 0.02. The total number of particles in the model $N_{tot}$ was set to $50V$ so that the number of protocells was approximately 100 irrespective of the value of $V$. At the beginning of each simulation, 50 protocells of equal size were generated. The initial values of $k_{pt}^c$ were set to $k_{max}$ for every replicator unless otherwise stated. The initial frequencies of P and Q were equal, and that of S was zero.

### (b) Ancestor tracking

Common ancestors of replicators were obtained in two steps. First, ancestor tracking was done at the cellular level to obtain the common ancestors of all surviving protocells. Second, ancestor tracking was done at the molecular level for the replicators contained by the common ancestors of protocells obtained in the first step. The results shown in figure 2e were obtained from the data between $2.1 \times 10^7$ and $2.17 \times 10^7$ time steps, so that the ancestor distribution was from after the completion of symmetry breaking.

## (c) Outline of the derivation of equations (3.1)

To derive equations (3.1), we simplified the agent-based model in two ways. First, we assumed that $k_{pt}^c$ is independent of $p$ and $t$. Under this assumption, a catalyst does not distinguish the replicator types of templates (i.e. $k_{pt}^c = k_{pt'}^c$ for $t \neq t'$) and products (i.e. $k_{pt}^c = k_{p't}^c$ for $p \neq p'$). This assumption excludes the possibility of numerical symmetry breaking, but still allows catalytic and informatic symmetry breaking as described in Results.

Second, we abstracted away chemical reactions by defining $\omega_{ij}^t$ as the probability that replicator $j$ of type $t$ in protocell $i$ is replicated or transcribed per unit time. Let $n_{ij}^t(\tau)$ be the population size of this replicator at time $\tau$. Then, $n_{ij}^t(\tau)$ is expected to satisfy

$$\begin{bmatrix} n_{ij}^P(\tau+1) \\ n_{ij}^Q(\tau+1) \end{bmatrix} = \begin{bmatrix} \omega_{ij}^P & \omega_{ij}^Q \\ \omega_{ij}^P & \omega_{ij}^Q \end{bmatrix} \begin{bmatrix} n_{ij}^P(\tau) \\ n_{ij}^Q(\tau) \end{bmatrix}. \tag{5.1}$$

The fitness of the replicator can be defined as the dominant eigenvalue $\lambda_{ij}$ of the $2 \times 2$ matrix on the right-hand side of equation (5.1): $\lambda_{ij} = \omega_{ij}^P + \omega_{ij}^Q$. Fisher's reproductive values of P and Q are given by the corresponding left eigenvector $\mathbf{u}_{ij} = [\omega_{ij}^P, \omega_{ij}^Q]$.

The evolutionary dynamics of the average catalytic activity of replicators can be described with Price's equation [8,9]. Let $\kappa_{ij}^c$ be the catalytic activity of replicator $j$ of type $c$ in protocell $i$ (we use $\kappa$ instead of $k$ to distinguish $\kappa_{ij}^c$ from $k_{pt}^c$). Price's equation states that

$$\langle \lambda_{i\tilde{j}} \rangle \Delta \langle \kappa_{\tilde{i}\tilde{i}}^c \rangle = \sigma_i^2[\langle \lambda_{i\tilde{j}} \rangle, \langle \kappa_{\tilde{i}j}^c \rangle] + \mathbb{E}_{\tilde{i}}[\sigma_{\tilde{i}j}^2[\lambda_{ij}, \kappa_{ij}^c]], \tag{5.2}$$

where $\langle x_{i\tilde{j}} \rangle$, $\langle x_{\tilde{i}j} \rangle$, and $\mathbb{E}_{\tilde{i}}[x]$ are $x$ averaged over the indices marked with tildes, $\sigma_i^2[x, y]$ is the covariance between $x$ and $y$ over protocells, and $\sigma_{\tilde{i}j}^2[x, y]$ is the covariance between $x$ and $y$ over the replicators in protocell $i$. One replicator is always counted as one sample in calculating all moments.

To approximate equation (5.2), we assumed that covariances between $\kappa_{ij}^P$ and $\kappa_{ij}^Q$ and between $\langle \kappa_{ij}^P \rangle$ and $\langle \kappa_{ij}^Q \rangle$ are negligible because the mutation of $\kappa_{ij}^P$ and that of $\kappa_{ij}^Q$ are uncorrelated in the agent-based model (see electronic supplementary material, Text 1.6 for an alternative justification of this assumption). Under this assumption, equation (5.2) is approximated by equations (3.1) up to the second central moments of $\kappa_{ij}^c$ and $\langle \kappa_{ij}^c \rangle$, with the following notation (see electronic supplementary material, Text 1.3 for the derivation):

$$\bar{\omega}^t = \frac{\langle \omega_{\tilde{i}\tilde{j}}^t \rangle}{\langle \lambda_{\tilde{i}\tilde{j}} \rangle}, \quad \sigma_{cel}^2 = \sigma_i^2[\langle \kappa_{\tilde{i}j}^c \rangle, \langle \kappa_{\tilde{i}j}^c \rangle], \quad \sigma_{mol}^2 = \mathbb{E}_{\tilde{i}}[\sigma_{\tilde{i}j}^2[\kappa_{ij}^c, \kappa_{ij}^c]],$$

$$\bar{k}^c = \langle \kappa_{\tilde{i}\tilde{j}}^c \rangle, \quad \gamma_c^t = -\mathbb{E}_{\tilde{i}}\left[\frac{\partial \ln \omega_{ij}^c}{\partial \kappa_{ij}^c}\right], \quad \beta_c^t = \frac{\partial \ln \langle \omega_{\tilde{i}\tilde{j}}^t \rangle}{\partial \langle \kappa_{\tilde{i}j}^c \rangle},$$

where $\bar{\omega}^t$ is the normalized average reproductive value of type-$t$ replicators, $\sigma_{cel}^2$, $\sigma_{mol}^2$, and $\bar{k}^c$ are the simplification of the notation, $\gamma_c^t$ is an average decrease in the replication rate of a type-$c$ replicator due to an increase in its own catalytic activity, and $\beta_c^t$ is an increase in the average replication rate of type-$t$ replicators in a protocell due to an increase in the average catalytic activity of type-$c$ replicators in that protocell. We assumed that $\sigma_{cel}^2$ and $\sigma_{mol}^2$ do not depend on $c$ because no difference is a priori assumed between P and Q.

The values of $\gamma_c^t$ and $\beta_c^t$ can be interpreted as the cost and benefit of providing catalysis. Let us assume that $V$ is so large that $\langle \kappa_{\tilde{i}j}^c \rangle$ and $\kappa_{ij}^c$ can be regarded as mathematically independent of each other if $i$ and $j$ are fixed (if $i$ and $j$ are varied, $\langle \kappa_{\tilde{i}j}^c \rangle$ and $\kappa_{ij}^c$

may be statistically correlated). Under this assumption, increasing $\kappa_{ij}^c$ does not increase $\langle \kappa_{ij}^c \rangle$, so that $\gamma_c^t$ reflects only the cost of providing catalysis at the molecular level. Likewise, increasing $\langle \kappa_{ij}^c \rangle$ does not increase $\kappa_{ij}^c$, so that $\beta_c^t$ reflects only the benefit of receiving catalysis at the cellular level. Moreover, the independence of $\langle \kappa_{ij}^c \rangle$ from $\kappa_{ij}^c$ implies that $\partial \omega_{ij}^{c'}/\partial \kappa_{ij}^c = 0$ for $c \neq c'$, which permits the following interpretation: if a replicator of type $c$ provides more catalysis, its transcripts, which is of type $c'$, pay no extra cost (i.e. $\gamma_c^{c'} = 0$).

## (d) Outline of the phase-plane analysis

To perform the phase-plane analysis depicted in figure 3, we defined $\omega_{ij}^t$ as a specific function of $\kappa_{ij}^t$ (see above for the meaning of $\omega_{ij}^t$ and $\kappa_{ij}^t$):

$$\omega_{ij}^t = e^{\langle \kappa_{ij}^P \rangle + \langle \kappa_{ij}^Q \rangle} e^{-s\kappa_{ij}^t}[\langle e^{-s\kappa_{ij}^P} \rangle + \langle e^{-s\kappa_{ij}^Q} \rangle]^{-1}, \tag{5.3}$$

where the first factor $e^{\langle \kappa_{ij}^P \rangle + \langle \kappa_{ij}^Q \rangle}$ represents the cellular-level benefit of catalysis provided by the replicators in protocell $i$, the second factor $e^{-s\kappa_{ij}^t}$ represents the molecular-level cost of catalysis provided by the focal replicator, the last factor normalizes the cost, and $s$ is the cost-benefit ratio. The above definition of $\omega_{ij}^t$ was chosen to satisfy the requirement that a replicator faces the trade-off between providing catalysis and serving as a template, i.e. $\gamma_t^t$ and $\beta_c^t$ are positive. Apart from this requirement, the definition was arbitrarily chosen for simplicity.

Under the definition in equation (5.3), we again approximated equation (5.2) up to the second central moments of $\kappa_{ij}^c$ and $\langle \kappa_{ij}^c \rangle$, obtaining the following (see electronic supplementary material, Text 1.6 for the derivation):

$$\bar{\omega}^t = \frac{e^{-s\bar{k}^t}}{e^{-s\bar{k}^P} + e^{-s\bar{k}^Q}}, \quad \gamma_c^t = s \quad \text{and} \quad \beta_c^t = 1. \tag{5.4}$$

Equations (3.1) and (5.4) can be expressed in a compact form as

$$\begin{bmatrix} \Delta \bar{k}^P \\ \Delta \bar{k}^Q \end{bmatrix} \approx \sigma_{tot}^2 \nabla[RB - (1-R)C],$$

where $\nabla = [\partial/\partial \bar{k}^P, \partial/\partial \bar{k}^Q]^T$ ($^T$ denotes transpose), $\sigma_{tot}^2 = \sigma_{mol}^2 + \sigma_{cel}^2$, $R = \sigma_{cel}^2/\sigma_{tot}^2$, $B = \bar{k}^P + \bar{k}^Q$ and $C = -\ln(e^{-s\bar{k}^P} + e^{-s\bar{k}^Q})$. $R$ can be interpreted as the regression coefficient of $\langle \kappa_{\tilde{i}j}^c \rangle$ on $\kappa_{ij}^c$ [39] and, therefore, the coefficient of genetic relatedness [40]. The potential $RB - (1-R)C$ can be interpreted as inclusive fitness.

Data accessibility. C++ source code implementing the agent-based model is available from the Dryad Digital Repository: https://doi.org/10.5061/dryad.mn257gm [41].

Authors' contributions. N.T. conceived the study, designed, implemented, and analysed the models, and wrote the paper. K.K. discussed the design, results, and implications of the study, and commented on the manuscript at all stages.

Competing interests. We declare we have no competing interests.

Funding. The authors have been supported by JSPS KAKENHI (grant nos JP17K17657 and JP17H06386). N.T. has been supported by grants from the University of Tokyo and the School of Biological Sciences, the University of Auckland.

Acknowledgements. The authors thank Stuart A. West and his group, and Ulrich F Müller for discussion, Daniel J. van der Post, Austen R. D. Ganley and Anthony M. Poole for help with the manuscript and Paulien Hogeweg for inspiration. The authors wish to acknowledge the contribution of NeSI to the results of this research. New Zealand's national compute and analytics services and team are supported by the New Zealand eScience Infrastructure (NeSI) and funded jointly by NeSI's collaborator institutions and through the Ministry of Business, Innovation and Employment. URL http://www.nesi.org.nz

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
