## [Reviewer comments · Proceedings of the Royal Society B: Biological Sciences]

Review History

RSPB-2019-0870.R0 (Original submission)

Review form: Reviewer 1 (Andy Gardner)

Recommendation

Major revision is needed (please make suggestions in comments)

Scientific importance: Is the manuscript an original and important contribution to its field?

Good

General interest: Is the paper of sufficient general interest?

Good

Quality of the paper: Is the overall quality of the paper suitable?

Poor

Is the length of the paper justified?

Yes

Should the paper be seen by a specialist statistical reviewer?

No

Do you have any concerns about statistical analyses in this paper? If so, please specify them explicitly in your report.

No

It is a condition of publication that authors make their supporting data, code and materials available - either as supplementary material or hosted in an external repository. Please rate, if applicable, the supporting data on the following criteria.

Is it accessible?

N/A

Is it clear?

N/A

Is it adequate?

N/A

Do you have any ethical concerns with this paper?

No

Comments to the Author

The purpose of this paper is to show that the Central Dogma of molecular biology – i.e. that there is (i) a separation of “genome” and “enzymes”, and (ii) a flow of information from “genome” to “enzymes” but not vice versa – arises as a basic consequence of multilevel selection. In this particular instance, the claim is that there is selection at the molecular level for higher replicative ability (which characterises the “genome”), and there is selection at the cellular level for higher catalytic ability (which characterises the “enzymes”), but the principle is intended to apply more generally, to any system of evolutionary replicators.

I think this is a really interesting topic, and that an analysis of this problem in these terms could make a really strong contribution to the literature. But, sadly, I didn't get very much out of reading this paper, on account of the vagueness of the model description, which made it really unclear what exactly was being assumed, and also on account of the lack of justification for the various assumptions, which made me uncertain as to what extent the results are biologically meaningful versus mathematical artefacts. This is a real shame, and I feel that with some extra care taken in its presentation, this paper might yet make a very strong contribution.

To give a concrete example: in line 76, we're told that the “total number of P, Q and S” is assumed to be constant. I initially thought this meant that the sum of n_P , n_Q and n_S is being held constant, but reading on I began to suspect that each of these quantities is supposed to be a constant in its own right. In addition to the vagueness, there's a lack of justification given for this assumption – and note that providing justification for an assumption would go a long way towards helping the reader understand exactly what is being assumed.

The assumption $n_P + n_Q + n_S = \text{constant}$ seems to me like it could be relatively benign, i.e. that it wouldn't interfere too much with the results. But the assumption $n_P = n_Q$ seems like it could have much more direct impact on the results, and if there's anything artificial about that assumption (in terms of a mismatch between the modelled scenario and the biological reality it is

supposed to describe) then that would make it relatively malignant and make the biological relevance of the results quite doubtful.

Perhaps in anticipation of this concern, the authors do refer to other versions of the model that lead robustly to the same overall conclusions (though no details are given in the main text). At the beginning of the supplementary material, the authors seem to be saying that they also considered a model where n_P and n_Q could vary, relative to each other. But here they say that the $n_P = n_Q$ assumption of the main text is due to cell-level selection, and that the model variant where n_P and n_Q are allowed to differ involves the absence of cell-level selection, so I'm confused as to how they can claim that multilevel selection drives the evolution of the Central Dogma situation if a variant model in which there is only selection at the lower level is supposed to give essentially the same outcome.

In short, the authors need to spend a bit more time outlining exactly what kind of scenarios they have in mind, and nailing down the biology a bit more firmly so that the reader can judge whether the model is capturing what it is supposed to.

A final point: the remarks about kin selection in lines 243-249 are a bit confused and confusing ("Kin selection theory posits that cooperation can evolve if genetic relatedness is sufficiently high", "for sufficiently low genetic relatedness... kin selection theory predicts that evolution cannot lead to cooperation; by contrast, our theory predicts that..."). The reality is that cooperation can be either altruistic or mutually beneficial, and that although genetic relatedness is required for altruism to be favoured it is not required for mutually beneficial cooperation to be favoured. West et al (2007, J Evol Biol 20:415) provides detailed discussion of the meaning of these various terms and their relation to kin selection theory.

As is my policy, I waive anonymity

Andy Gardner

Review form: Reviewer 2

Recommendation

Major revision is needed (please make suggestions in comments)

Scientific importance: Is the manuscript an original and important contribution to its field?

Good

General interest: Is the paper of sufficient general interest?

Good

Quality of the paper: Is the overall quality of the paper suitable?

Good

Is the length of the paper justified?

Yes

Should the paper be seen by a specialist statistical reviewer?

No

Do you have any concerns about statistical analyses in this paper? If so, please specify them explicitly in your report.

No

It is a condition of publication that authors make their supporting data, code and materials available - either as supplementary material or hosted in an external repository. Please rate, if applicable, the supporting data on the following criteria.

Is it accessible?

N/A

Is it clear?

N/A

Is it adequate?

N/A

Do you have any ethical concerns with this paper?

No

Comments to the Author

Review of central dogma paper

This paper addresses a fundamental problem in biology: the origin of the genotype and phenotype. The paper casts this as the problem of the origin of the central dogma, which is fine, but the problem is bigger than that. The paper shows that with appropriate trade-offs between two fitness components, template replication and catalysis, specialization in each can occur. I like the paper and think the results are interesting and should be published. There are two areas that need to be addressed mainly as background for the present paper.

The first is the theory of hypercycles. Fig. 1a is a two-member hypercycle. These have been analyzed by a variety of authors. Why is the present paper not building on that extensive body of work? Why use an agent-based model when difference or differential equations have been used previously to study similar problems?

The second issue concerns a paper by Michod referenced below which basically asks the same question as the present paper. This 1983 paper is extended in Chapters 2 and 3 of reference [5] in the present paper (see especially Fig 3-2 and related text). The present paper should clarify what are the similarities and differences in approach and results between this previous work and the present paper.

R. E. Michod, Population biology of the first replicators: on the origin of the genotype, phenotype and organism. *Am. Zool.* 23, 5-14 (1983).

Decision letter (RSPB-2019-0870.R0)

20-May-2019

Dear Dr Takeuchi:

I am writing to inform you that your manuscript RSPB-2019-0870 entitled "The origin of the

central dogma through conflicting multilevel selection" has, in its current form, been rejected for publication in Proceedings B.

This action has been taken on the advice of referees, who have recommended that substantial revisions are necessary. With this in mind we would be happy to consider a resubmission, provided the comments of the referees are fully addressed. However please note that this is not a provisional acceptance.

Sincerely,
Proceedings B
mailto: proceedingsb@royalsociety.org

Associate Editor
Board Member: 1
Comments to Author:

If in the primal soup of molecules both capable of storing information and doing catalytic action, what evolutionary processes gave rise to the strict separation we observe today where nucleic acids store information and protein do things? Apart from the fact that this separation is not always so clear cut as often thought, it is a valid question to ask how the so-called fundamental dogma arose. This MS claims to show that it can arise of a combination of an information-storage / replication constraint and multilevel selection.

Both reviewers agree that the MS deals with an interesting topic, but both express a number of criticisms. One reviewer argues that the MS should be more justice to previous attempts (from the 1980s!) to address this problem. The other reviewer (Gardner), however, has serious problems with the way the model is described and I agree that the model description could be much clearer. I thus think the MS would significantly improve if his suggestions were incorporated.

Reviewer(s)' Comments to Author:

Referee: 1

Comments to the Author(s)

The purpose of this paper is to show that the Central Dogma of molecular biology – i.e. that there is (i) a separation of “genome” and “enzymes”, and (ii) a flow of information from “genome” to “enzymes” but not vice versa – arises as a basic consequence of multilevel selection. In this particular instance, the claim is that there is selection at the molecular level for higher replicative ability (which characterises the “genome”), and there is selection at the cellular level for higher catalytic ability (which characterises the “enzymes”), but the principle is intended to apply more generally, to any system of evolutionary replicators.

I think this is a really interesting topic, and that an analysis of this problem in these terms could make a really strong contribution to the literature. But, sadly, I didn't get very much out of reading this paper, on account of the vagueness of the model description, which made it really unclear what exactly was being assumed, and also on account of the lack of justification for the various assumptions, which made me uncertain as to what extent the results are biologically meaningful versus mathematical artefacts. This is a real shame, and I feel that with some extra care taken in its presentation, this paper might yet make a very strong contribution.

To give a concrete example: in line 76, we're told that the “total number of P, Q and S” is assumed to be constant. I initially thought this meant that the sum of n_P , n_Q and n_S is being held constant, but reading on I began to suspect that each of these quantities is supposed to be a constant in its own right. In addition to the vagueness, there's a lack of justification given for this assumption – and note that providing justification for an assumption would go a long way towards helping the reader understand exactly what is being assumed.

The assumption $n_P + n_Q + n_S = \text{constant}$ seems to me like it could be relatively benign, i.e. that it wouldn't interfere too much with the results. But the assumption $n_P = n_Q$ seems like it could have much more direct impact on the results, and if there's anything artificial about that assumption (in terms of a mismatch between the modelled scenario and the biological reality it is supposed to describe) then that would make it relatively malignant and make the biological relevance of the results quite doubtful.

Perhaps in anticipation of this concern, the authors do refer to other versions of the model that lead robustly to the same overall conclusions (though no details are given in the main text). At the beginning of the supplementary material, the authors seem to be saying that they also considered a model where n_P and n_Q could vary, relative to each other. But here they say that the $n_P = n_Q$ assumption of the main text is due to cell-level selection, and that the model variant where n_P and n_Q are allowed to differ involves the absence of cell-level selection, so I'm confused as to how they can claim that multilevel selection drives the evolution of the Central Dogma situation if a variant model in which there is only selection at the lower level is supposed to give essentially the same outcome.

In short, the authors need to spend a bit more time outlining exactly what kind of scenarios they have in mind, and nailing down the biology a bit more firmly so that the reader can judge whether the model is capturing what it is supposed to.

A final point: the remarks about kin selection in lines 243-249 are a bit confused and confusing (“Kin selection theory posits that cooperation can evolve if genetic relatedness is sufficiently high”, “for sufficiently low genetic relatedness... kin selection theory predicts that evolution cannot lead to cooperation; by contrast, our theory predicts that...”). The reality is that

cooperation can be either altruistic or mutually beneficial, and that although genetic relatedness is required for altruism to be favoured it is not required for mutually beneficial cooperation to be favoured. West et al (2007, *J Evol Biol* 20:415) provides detailed discussion of the meaning of these various terms and their relation to kin selection theory.

As is my policy, I waive anonymity

Andy Gardner

Referee: 2

Comments to the Author(s)
Review of central dogma paper

This paper addresses a fundamental problem in biology: the origin of the genotype and phenotype. The paper casts this as the problem of the origin of the central dogma, which is fine, but the problem is bigger than that. The paper shows that with appropriate trade-offs between two fitness components, template replication and catalysis, specialization in each can occur. I like the paper and think the results are interesting and should be published. There are two areas that need to be addressed mainly as background for the present paper.

The first is the theory of hypercycles. Fig. 1a is a two-member hypercycle. These have been analyzed by a variety of authors. Why is the present paper not building on that extensive body of work? Why use an agent-based model when difference or differential equations have been used previously to study similar problems?

The second issue concerns a paper by Michod referenced below which basically asks the same question as the present paper. This 1983 paper is extended in Chapters 2 and 3 of reference [5] in the present paper (see especially Fig 3-2 and related text). The present paper should clarify what are the similarities and differences in approach and results between this previous work and the present paper.

R. E. Michod, Population biology of the first replicators: on the origin of the genotype, phenotype and organism. *Am. Zool.* 23, 5-14 (1983).

Author's Response to Decision Letter for (RSPB-2019-0870.R0)

See Appendix A.

RSPB-2019-1359.R0

Review form: Reviewer 2 (Rick Michod)

Recommendation

Major revision is needed (please make suggestions in comments)

Scientific importance: Is the manuscript an original and important contribution to its field?

Excellent

General interest: Is the paper of sufficient general interest?

Excellent

Quality of the paper: Is the overall quality of the paper suitable?

Acceptable

Is the length of the paper justified?

Yes

Should the paper be seen by a specialist statistical reviewer?

No

Do you have any concerns about statistical analyses in this paper? If so, please specify them explicitly in your report.

No

It is a condition of publication that authors make their supporting data, code and materials available - either as supplementary material or hosted in an external repository. Please rate, if applicable, the supporting data on the following criteria.

Is it accessible?

N/A

Is it clear?

N/A

Is it adequate?

N/A

Do you have any ethical concerns with this paper?

No

Comments to the Author

I accept the author's reply to my first point. I did not appreciate that P and Q are templates for the other in addition to being catalytic.

I do not feel the authors have addressed my second point copied below.

"The second issue concerns a paper by Michod referenced below which basically asks the same question as the present paper. This paper is extended in Chapters 2 and 3 of reference [5] in the present paper (see especially Fig 3-2 and related text). The present paper should clarify what are the similarities and differences in approach and results. R. E. Michod, Population biology of the

first replicators: on the origin of the genotype, phenotype and organism. *Am. Zool.* 23, 5–14 (1983). “

In that 1983 paper, I tried to address the same general question being addressed in the present paper, how does the phenotype diverge from the genotype. I did this in a qualitative way, the present paper goes beyond that with an explicit model. Still I said similar things. I said things like: “Thus, with replication, natural selection and life-history evolution began. The genotype, or nucleotide sequence, of the replicating molecule gave rise to several phenotypic properties, the most important of which was its three-dimensional structure which in turn affected the birth and death processes. However, at this stage of nonenzymatic template replication, the phenotype was the physical structure of the genotype, nothing more; For the divergence of the phenotype from the genotype it was necessary for the replicator to produce a protein. It is shown here that the evolution of enzyme production is facilitated by the existence of population structure in the distribution of the macromolecules associated with replication.” I made the point about the importance of trade-offs in fitness components concluding “Thus there are adaptive constraints on the birth and death process even at this primitive molecular level.” Or in Chapter 2 of my book where the same issues are discussed “The molecular replicator without making proteins would likely encounter constraints in simultaneously maximizing birth and survival. A more open structure facilitates birth but risks death in a hydrolytic environment.” My guess is that symmetry breaking in the present paper is some kind of covariance effect as discussed in my 2006 PNAS paper with the title “The group covariance effect and fitness trade-offs during evolutionary transitions in individuality”.

My point here is only that an appreciation of the history of the question and previous attempts to answer it should be given in the present paper. It is fine if the authors don't agree with my 1983 analysis, or find it limited, or different from theirs. Indeed, I find their analysis more satisfying. But some discussion should be given and they should clarify what the similarities and differences are in the two analyses with regard to answering the question of how the phenotype diverged from the genotype.

The author's say in their rebuttal. “Given that Michod's model is concerned with the origin of hypercycles, we feel that a comparison with his model is implicitly included in the new paragraphs mentioned above. If we missed something, we would be glad to know about it.”

To just say one (Michod 1983) is based on hypercycles and their analysis isn't doesn't go to the crux of the issue.

Here is what I think they have missed. There are two fitness components for a replicator, its capacity to encode information and its capacity to copy or replicate that information. The crux of the issue is simply that a replicator without an enzyme cannot simultaneously optimize both its information carrying capacity and the functional capacities that promote replication, when there are trade-offs involved between these two fitness components. By making an enzyme, the phenotype (functional capacities) can diverge from the genotype (information carrying capacity) and both can be optimized or enhanced independently. That is the essence of the model results (of course there is more) and the basic point I made in 1983.

Decision letter (RSPB-2019-1359.R0)

27-Jun-2019

Dear Dr Takeuchi:

Your manuscript has now been peer reviewed and the reviews have been assessed by an Associate Editor. The reviewer's comments (not including confidential comments to the Editor) and the comments from the Associate Editor are included at the end of this email for your reference. As you will see, the reviewer has raised some issues with your manuscript and we would like to invite you to revise your manuscript to address them.

Research ethics:

Use of animals and field studies:

It is a condition of publication that you make available the data and research materials supporting the results in the article. Datasets should be deposited in an appropriate publicly available repository and details of the associated accession number, link or DOI to the datasets must be included in the Data Accessibility section of the article

(<https://royalsociety.org/journals/ethics-policies/data-sharing-mining/>). Reference(s) to datasets should also be included in the reference list of the article with DOIs (where available).

Please submit a copy of your revised paper within three weeks. If we do not hear from you within this time your manuscript will be rejected. If you are unable to meet this deadline please let us know as soon as possible, as we may be able to grant a short extension.

Best wishes,
Professor Hans Heesterbeek
mailto: proceedingsb@royalsociety.org

Associate Editor Board Member

Comments to Author:

I think it shouldn't be to difficult to deal with the referee's second point

Reviewer(s)' Comments to Author:

Referee: 2

Comments to the Author(s).

I accept the author's reply to my first point. I did not appreciate that P and Q are templates for the other in addition to being catalytic.

I do not feel the authors have addressed my second point copied below.

“The second issue concerns a paper by Michod referenced below which basically asks the same question as the present paper. This paper is extended in Chapters 2 and 3 of reference [5] in the present paper (see especially Fig 3-2 and related text). The present paper should clarify what are the similarities and differences in approach and results. R. E. Michod, Population biology of the first replicators: on the origin of the genotype, phenotype and organism. *Am. Zool.* 23, 5-14 (1983).”

In that 1983 paper, I tried to address the same general question being addressed in the present paper, how does the phenotype diverge from the genotype. I did this in a qualitative way, the present paper goes beyond that with an explicit model. Still I said similar things. I said things like: “Thus, with replication, natural selection and life-history evolution began. The genotype, or nucleotide sequence, of the replicating molecule gave rise to several phenotypic properties, the most important of which was its three-dimensional structure which in turn affected the birth and death processes. However, at this stage of nonenzymatic template replication, the phenotype was the physical structure of the genotype, nothing more; For the divergence of the phenotype from the genotype it was necessary for the replicator to produce a protein. It is shown here that the evolution of enzyme production is facilitated by the existence of population structure in the distribution of the macromolecules associated with replication.” I made the point about the importance of trade-offs in fitness components concluding “Thus there are adaptive constraints on the birth and death process even at this primitive molecular level.” Or in Chapter 2 of my book where the same issues are discussed “The molecular replicator without making proteins would likely encounter constraints in simultaneously maximizing birth and survival. A more open structure facilitates birth but risks death in a hydrolytic environment.” My guess is that symmetry breaking in the present paper is some kind of covariance effect as discussed in my 2006 PNAS paper with the title “The group covariance effect and fitness trade-offs during evolutionary transitions in individuality”.

My point here is only that an appreciation of the history of the question and previous attempts to answer it should be given in the present paper. It is fine if the authors don't agree with my 1983 analysis, or find it limited, or different from theirs. Indeed, I find their analysis more satisfying. But some discussion should be given and they should clarify what the similarities and differences are in the two analyses with regard to answering the question of how the phenotype diverged from the genotype.

The author's say in their rebuttal. “Given that Michod's model is concerned with the origin of hypercycles, we feel that a comparison with his model is implicitly included in the new paragraphs mentioned above. If we missed something, we would be glad to know about it.”

To just say one (Michod 1983) is based on hypercycles and their analysis isn't doesn't go to the crux of the issue.

Here is what I think they have missed. There are two fitness components for a replicator, its capacity to encode information and its capacity to copy or replicate that information. The crux of the issue is simply that a replicator without an enzyme cannot simultaneously optimize both its information carrying capacity and the functional capacities that promote replication, when there are trade-offs involved between these two fitness components. By making an enzyme, the phenotype (functional capacities) can diverge from the genotype (information carrying capacity) and both can be optimized or enhanced independently. That is the essence of the model results (of course there is more) and the basic point I made in 1983.

Author's Response to Decision Letter for (RSPB-2019-1359.R0)

See Appendix B.

RSPB-2019-1359.R1 (Revision)

Review form: Reviewer 2 (Rick Michod)

Recommendation

Major revision is needed (please make suggestions in comments)

Scientific importance: Is the manuscript an original and important contribution to its field?

Good

General interest: Is the paper of sufficient general interest?

Good

Quality of the paper: Is the overall quality of the paper suitable?

Marginal

Is the length of the paper justified?

Yes

Should the paper be seen by a specialist statistical reviewer?

No

Do you have any concerns about statistical analyses in this paper? If so, please specify them explicitly in your report.

No

It is a condition of publication that authors make their supporting data, code and materials available - either as supplementary material or hosted in an external repository. Please rate, if applicable, the supporting data on the following criteria.

Is it accessible?

N/A

Is it clear?

N/A

Is it adequate?

N/A

Do you have any ethical concerns with this paper?

Yes

Comments to the Author

See attached . (See Appendix C)

Decision letter (RSPB-2019-1359.R1)

26-Jul-2019

Dear Dr Takeuchi:

Your manuscript has now been peer reviewed and the reviews have been assessed by an Associate Editor. The reviewer's comments (not including confidential comments to the Editor) are attached and the comments from the Associate Editor are included at the end of this email for your reference. As you will see, a crucial issue remains to be settled. We clearly feel there is important overlap between the Michod study from 1983 and your study. The study must therefore be discussed in this way and suitably given the credited it is due. Simply quoting it in the reference list will not do. I will deviate from our policy and give you the opportunity to revise a final time. Let it be clear though that the manuscript will be rejected in the next round if this issue is not satisfactorily resolved.

Research ethics:

Use of animals and field studies:

If you wish to submit your data to Dryad (<http://datadryad.org/>) and have not already done so you can submit your data via this link [http://datadryad.org/submit?journalID=RSPB&manu=\(Document not available\)](http://datadryad.org/submit?journalID=RSPB&manu=(Document%20not%20available)), which will take you to your unique entry in the Dryad repository.

Please submit a copy of your revised paper within three weeks. If we do not hear from you within this time your manuscript will be rejected. If you are unable to meet this deadline please let us know as soon as possible, as we may be able to grant a short extension.

Best wishes,
Professor Hans Heesterbeek
Editor, Proceedings B
mailto: proceedingsb@royalsociety.org

Associate Editor

Comments to Author:

As perhaps could be expected, the reviewer doesn't subscribe to your conclusion that his 1983 paper is about something different, and I agree with him. One cannot say that the current model

is different from Michod's because it depends on conflicting levels of selection whereas Michod's 1983 analysis is based on kin selection. Maybe there is a subtle issue here, but in my mind kin and group selection were invented precisely to be able to study the consequences of conflicting levels of selection! I think the MS cannot be accepted without crediting Michod's 1983 study.

Reviewer(s)' Comments to Author:

Referee: 2

Comments to the Author(s)
See attached

Author's Response to Decision Letter for (RSPB-2019-1359.R1)

See Appendix D.

Decision letter (RSPB-2019-1359.R2)

21-Aug-2019

Dear Dr Takeuchi:

Your manuscript has now been reviewed by the Associate Editor. Please find his comments at the end of this email. As you will see, there is one final issue that needs to be dealt with. I agree with the Associate Editor and his arguments are compelling. This manuscript has now gone through several more revisions than we usually allow. Please be aware that the next one is the final revision. If the issue is not solved in a way that fully satisfies the Associate Editor, I will reject the manuscript. Perhaps this clarity gives us the convergence we need.

When revising your manuscript you should also ensure that it adheres to our editorial policies

(<https://royalsociety.org/journals/ethics-policies/>). You should pay particular attention to the following:

Research ethics:

Use of animals and field studies:

Please submit a copy of your revised paper within three weeks. If we do not hear from you within this time your manuscript will be rejected. If you are unable to meet this deadline please let us know as soon as possible, as we may be able to grant a short extension.

Best wishes,
Professor Hans Heesterbeek
Editor, Proceedings B
mailto: proceedingsb@royalsociety.org

Associate Editor
Comments to Author:

This MS has gone through a few rounds already but seems to be stuck in a standoff between the authors and a reviewer, Rick Michod. I was quite reluctant to send it yet again to the reviewer and decided to see whether I fully understood what the standoff is about. Michod claims that this MS addresses the same question as one of his early (1983) paper; the authors feel this paper deals with something different. They did cite a later paper by Michod which is indeed relevant but this does not address the same question (the reference has gone from the present version). Looking up Michod (1983), I have to conclude that it indeed addresses a very similar question as the present MS:

"The purpose of this article is to study the selective pressures responsible for the divergence of the phenotype from the genotype, and the origin of the cellular organism."

The way the authors deal with it is indeed rather different and, as also Michod concedes, goes to a more fundamental level. As the authors note in their rebuttal letter, Michod (1983) studies various hypercycle models which takes the existence of separate catalysers for granted whereas the present MS allows these to emerge by symmetry breaking. I consider these results novel and worthy of publication in the Proceedings. However, I still have trouble with the way they, reluctantly it seems, deal with Michod's early paper.

In the preceding back-and-forth the authors claimed the main difference between the two approaches was that Michod's results depend on kin selection whereas their finding results from "conflicting multilevel selection". I am aware that there is some controversy regarding these mechanisms but for the present issue this is irrelevant: though Michod discusses his results in kin selection terms, the model he analyses is a proper group selection (= multilevel) model.

In the last revision the authors avoided the kin selection-multilevel selection issue, which is very welcome, and added a discussion of Michod's work which is also nice. Although I do not fully agree with all they claim, to me this is not really a problem, considering it is up to them to defend their views if it stirs up a polemic. However, Michod's paper is only the 36th to be cited, toward the very end of the discussion, and I think it should be mentioned much earlier.

Although I do like the MS and the analysis it presents and I do think it will have a significant impact if published, I still hesitate to recommend it. If the authors acknowledge in the introduction that the question has been addressed before but that their approach delivers some novel insights all would be fine with me.

Author's Response to Decision Letter for (RSPB-2019-1359.R2)

See Appendix E.

Decision letter (RSPB-2019-1359.R3)

09-Sep-2019

Dear Dr Takeuchi

I am pleased to inform you that your manuscript entitled "The origin of the central dogma through conflicting multilevel selection" has been accepted for publication in Proceedings B.

Open Access

Paper charges

Sincerely,

Professor Hans Heesterbeek
Editor, Proceedings B
<mailto:proceedingsb@royalsociety.org>

Appendix A

(The comments of the editors and reviewers are italicised)

Reply to Board Member 1

If in the primal soup of molecules both capable of storing information and doing catalytic action, what evolutionary processes gave rise to the strict separation we observe today where nucleic acids store information and protein do things? Apart from the fact that this separation is not always so clear cut as often thought, it is a valid question to ask how the so-called fundamental dogma arose. This MS claims to show that it can arise from a combination of an information-storage / replication constraint and multilevel selection.

Both reviewers agree that the MS deals with an interesting topic, but both express a number of criticisms. One reviewer argues that the MS should be more justice to previous attempts (from the 1980s!) to address this problem. The other reviewer (Gardner), however, has serious problems with the way the model is described and I agree that the model description could be much clearer. I thus think the MS would significantly improve if his suggestions were incorporated.

We thank the editor and referees for considering our manuscript. We are pleased with the first paragraphs of the editor and referees' comments, which summarise the motivation and essence of our work in perceptive manners. We have revised the manuscript based on the referees' comments as detailed below. We are keen to know what the referees think of the outcomes of our revision.

Reply to Referee 1's comments:

We have revised the manuscript based on the referee's comments as described below. In this revised submission, we are providing two versions of an identical manuscript, with and without all revisions highlighted. The page and line numbers shown below are for the version with the highlighting.

The purpose of this paper is to show that the Central Dogma of molecular biology – i.e. that there is (i) a separation of “genome” and “enzymes”, and (ii) a flow of information from “genome” to “enzymes” but not vice versa – arises as a basic consequence of multilevel selection. In this particular instance, the claim is that there is selection at the

molecular level for higher replicative ability (which characterises the “genome”), and there is selection at the cellular level for higher catalytic ability (which characterises the “enzymes”), but the principle is intended to apply more generally, to any system of evolutionary replicators.

I think this is a really interesting topic, and that an analysis of this problem in these terms could make a really strong contribution to the literature. But, sadly, I didn't get very much out of reading this paper, on account of the vagueness of the model description, which made it really unclear what exactly was being assumed, and also on account of the lack of justification for the various assumptions, which made me uncertain as to what extent the results are biologically meaningful versus mathematical artefacts. This is a real shame, and I feel that with some extra care taken in its presentation, this paper might yet make a very strong contribution.

To give a concrete example: in line 76, we're told that the “total number of P, Q and S” is assumed to be constant. I initially thought this meant that the sum of n_P , n_Q and n_S is being held constant, but reading on I began to suspect that each of these quantities is supposed to be a constant in its own right. In addition to the vagueness, there's a lack of justification given for this assumption – and note that providing justification for an assumption would go a long way towards helping the reader understand exactly what is being assumed.

In all our agent-based models, the sum of n_P , n_Q and n_S is held constant, while the relative frequencies of P, Q, and S are variable. To minimise vagueness, we have revised Methods, which now states, "the total number of particles, i.e., the sum of the total numbers of P, Q, and S, is kept constant (the relative frequencies of P, Q, and S are variable)" (Page 3 Line 80-82).

While all our agent-based models do not assume $n_P = n_Q$, equations (1) do make this assumption for a reason described next. We have also provided the requested justifications as described later. The assumption $n_P + n_Q + n_S = \text{constant}$ seems to me like it could be relatively benign, i.e. that it wouldn't interfere too much with the results. But the assumption $n_P = n_Q$ seems like it could have much more direct

impact on the results, and if there's anything artificial about that assumption (in terms of a mismatch between the modelled scenario and the biological reality it is supposed to describe) then that would make it relatively malignant and make the biological relevance of the results quite doubtful.

The goal of our mathematical analysis was to understand the mechanism of catalytic and informatic symmetry breaking as displayed by our agent-based model. To simplify this goal, we focused our attention to catalytic and informatic symmetry breaking, setting aside numerical symmetry breaking for a moment. We thus assumed $n_P = n_Q$ for simplicity, and this assumption enabled us to derive equations (1). Using equations (1), we identified the mechanisms of catalytic and informatic symmetry breaking as described in the manuscript. Moreover, the analysis of equations (1) implies that catalytic and informatic symmetry breaking ensues even under the assumption of $n_P = n_Q$. Having dealt with catalytic and informatic symmetry breaking, we next turned our attention to numerical symmetry breaking. To identify the mechanism of numerical symmetry breaking, we used different equations, which relax the assumption of $n_P = n_Q$ (this is described in Supporting Text 1.4 [formerly 1.3]). Our analysis indicates that numerical symmetry breaking is a consequence, rather than a cause, of catalytic and informatic symmetry breaking. In summary, the assumption of $n_P = n_Q$ is made only for the sake of simplicity, and it is relaxed in the supplementary material.

To minimise the chance of confusing the reader regarding the above matter, we have revised our manuscript in two ways. First, we split Results into two sub-sections, titled "computational analysis" and "mathematical analysis", to indicate the separation of the texts describing the agent-based model and those describing the mathematical analysis of it.

The second revision involves the addition of justification for the simplifying assumptions made in the derivation of equations (1), in particular, for the assumption of $n_P = n_Q$. The revised manuscript now states that this assumption is made in equations (1) (but not in the agent-based model) to simplify our investigation into the mechanism of catalytic and informatic symmetry breaking (Page 8 Line 172-180). Immediately after this justification, the revised manuscript refers to the supplementary material

describing the mathematical analysis of numerical symmetry breaking in order to imply that we relax the assumption of $n_P = n_Q$ in a separate account (Page 8-9 Line 184-190). In the previous version of the manuscript, a reference to the same material was made much earlier where the agent-based model is described. This reference in an earlier location was removed to minimise the chance of confusion. Accordingly, the order of supplementary material texts was modified.

Perhaps in anticipation of this concern, the authors do refer to other versions of the model that lead robustly to the same overall conclusions (though no details are given in the main text). At the beginning of the supplementary material, the authors seem to be saying that they also considered a model where n_P and n_Q could vary, relative to each other. But here they say that the $n_P = n_Q$ assumption of the main text is due to cell-level selection, and that the model variant where n_P and n_Q are allowed to differ involves the absence of cell-level selection, so I'm confused as to how they can claim that multilevel selection drives the evolution of the Central Dogma situation if a variant model in which there is only selection at the lower level is supposed to give essentially the same outcome.

Both in the main text and at the beginning of the supplementary material, our agent-based models assume that the frequencies of P and Q are evolvable. Moreover, cellular-level selection is always present, tending to maximise the multiplication of replicators within protocells. The difference between the two versions of model lies in whether or not cellular-level selection favours $n_P = n_Q$. More specifically, the two versions differ in the conditions under which the multiplication of replicators is maximised. In the model described in the main text, replicators multiply fastest if P and Q coexist (in particular, $n_P = n_Q$); therefore, cellular-level selection favours coexistence between P and Q. By contrast, in the model described in the supplementary material, replicators multiply fastest even if either P or Q is totally absent; therefore, cellular-level selection does not necessarily favour coexistence between P and Q. The purpose of considering the two versions of model was to examine whether three-fold symmetry breaking is affected by the fact that cellular-level selection favours coexistence between P and Q. The results described in the supplementary material show that it is not affected. In fact, three-fold symmetry breaking occurs even if a system

initially contains only one type of replicator.

To minimise the chance of confusing the reader regarding the above matter, we revised the beginning part of the supplementary material (Supporting Texts 1.1). Besides revising the existing texts, we added the following specific points: (1) cellular-level selection always tends to maximise the multiplication rate of replicators within protocells; (2) cellular-level selection is indifferent to how this maximisation is achieved; (3) this maximisation is achieved under different conditions in the two versions of model.

In short, the authors need to spend a bit more time outlining exactly what kind of scenarios they have in mind, and nailing down the biology a bit more firmly so that the reader can judge whether the model is capturing what it is supposed to.

We have revised our manuscript based on the points raised by the referee as described above. If the referee realises any other specific points that we could improve upon, we would be keen to be made aware of them.

A final point: the remarks about kin selection in lines 243-249 are a bit confused and confusing (“Kin selection theory posits that cooperation can evolve if genetic relatedness is sufficiently high”, “for sufficiently low genetic relatedness... kin selection theory predicts that evolution cannot lead to cooperation; by contrast, our theory predicts that...”). The reality is that cooperation can be either altruistic or mutually beneficial, and that although genetic relatedness is required for altruism to be favoured it is not required for mutually beneficial cooperation to be favoured. West et al (2007, J Evol Biol 20:415) provides detailed discussion of the meaning of these various terms and their relation to kin selection theory.

Based on this comment, we replaced 'cooperation' with 'altruism.' To elaborate on this terminology, we also added the following sentence in brackets: 'providing catalysis can be viewed as altruism (West et al 2007): 'providing catalysis brings no direct benefit to a catalyst because a catalyst cannot catalyse the replication of itself in our model' (Page 13 Line 292-304).

Reply to Referee 2's comments:

We have revised the manuscript based on the referee's comments as described below. In this revised submission, we are providing two versions of an identical manuscript, with and without all revisions highlighted. The page and line numbers shown below are for the version with the highlighting.

This paper addresses a fundamental problem in biology: the origin of the genotype and phenotype. The paper casts this as the problem of the origin of the central dogma, which is fine, but the problem is bigger than that. The paper shows that with appropriate trade-offs between two fitness components, template replication and catalysis, specialization in each can occur. I like the paper and think the results are interesting and should be published. There are two areas that need to be addressed mainly as background for the present paper.

The first is the theory of hypercycles. Fig. 1a is a two-member hypercycle. These have been analysed by a variety of authors. Why is the present paper not building on that extensive body of work? Why use an agent-based model when difference or differential equations have been used previously to study similar problems?

Fig. 1a is not a two-member hypercycle because transcription occurs between P and Q. We added this remark to the main text.

Based on the referee's comment, we added three new paragraphs to Discussion comparing our theory with hypercycle theory. In our opinion, an important difference between the two theories lies not so much in the methods of modelling as in the proposed mechanisms for the evolution of complexity in replicator systems. Briefly, hypercycle theory proposes symbiosis between multiple lineages of replicators, whereas our theory proposes symmetry breaking (i.e., differentiation) in a single lineage of replicators, a fundamental distinction that is drawn between egalitarian and fraternal major evolutionary transitions as defined by Queller (please see Page 12, Line 267-276 for details). In addition, our theory differs from hypercycle theory in terms of the roles played by non-catalytic templates. In hypercycle theory, the evolution of such templates

(i.e., parasites) engenders hypercycles; in our theory, it is one of the essential factors leading to the division of labour between genomes and enzymes (please see Page 12, Line 277-282). Although our theory differs from hypercycle theory in these aspects, it does not contradict the latter. In fact, there is a potential synergy between the evolution of complexity via symbiosis and that via symmetry breaking (Page 12-13, Line 283-291).

We consider the above comparison important additions to the manuscript and thank Reviewer for leading us to make it.

The second issue concerns a paper by Michod referenced below which basically asks the same question as the present paper. This 1983 paper is extended in Chapters 2 and 3 of reference [5] in the present paper (see especially Fig 3-2 and related text). The present paper should clarify what are the similarities and differences in approach and results between this previous work and the present paper.

Michod's model in his 1983 paper, as well as its extension in his book, is concerned with hypercycles. In particular, Fig 3-2 in his book depicts conditions under which a hypercycle can invade or be maintained in the face of competition from a non-hypercycle, i.e., templates replicating in a non-enzymatic manner. Given that Michod's model is concerned with the origin of hypercycles, we feel that a comparison with his model is implicitly included in the new paragraphs mentioned above. If we missed something, we would be glad to know about it.

We would like to add that Michod's book also describes models for the origin of germ-soma distinction in multicellular organisms. It is tantalising to compare the germ-soma distinction displayed by his models to the genome-enzyme distinction displayed by our model. However, we have decided to refrain from this comparison, limiting our discussion along this line to those described in the last paragraph of our manuscript. We are currently generalising our protocell model so that multicellular organisms are more evidently within the scope of our model. Thus, we intend to refer to Michod's work more explicitly in the future.

Appendix B

(The comments of the reviewer are italicised)

Reply to Referee 2's comments:

We have revised the manuscript based on the referee's comments as described below. In this submission, we are again providing two versions of an identical manuscript, with and without all revisions highlighted. The page and line numbers shown below are for the version with the highlighting.

I accept the author's reply to my first point. I did not appreciate that P and Q are templates for the other in addition to being catalytic.

I do not feel the authors have addressed my second point copied below.

“The second issue concerns a paper by Michod referenced below which basically asks the same question as the present paper. This paper is extended in Chapters 2 and 3 of reference [5] in the present paper (see especially Fig 3-2 and related text). The present paper should clarify what are the similarities and differences in approach and results. R. E. Michod, Population biology of the first replicators: on the origin of the genotype, phenotype and organism. Am. Zool. 23, 5–14 (1983). “

In that 1983 paper, I tried to address the same general question being addressed in the present paper, how does the phenotype diverge from the genotype. I did this in a qualitative way, the present paper goes beyond that with an explicit model. Still I said similar things. I said things like: “Thus, with replication, natural selection and life-history evolution began. The genotype, or nucleotide sequence, of the replicating molecule gave rise to several phenotypic properties, the most important of which was its three-dimensional structure which in turn affected the birth and death processes. However, at this stage of nonenzymatic template replication, the phenotype was the physical structure of the genotype, nothing more; For the divergence of the phenotype from the genotype it was necessary for the replicator to produce a protein. It is shown here that the evolution of enzyme production is facilitated by the existence of population structure in the distribution of the macromolecules associated with replication.” I made the point about the importance of trade-offs in fitness components concluding “Thus

there are adaptive constraints on the birth and death process even at this primitive molecular level.” Or in Chapter 2 of my book where the same issues are discussed “The molecular replicator without making proteins would likely encounter constraints in simultaneously maximizing birth and survival. A more open structure facilitates birth but risks death in a hydrolytic environment.” My guess is that symmetry breaking in the present paper is some kind of covariance effect as discussed in my 2006 PNAS paper with the title “The group covariance effect and fitness trade-offs during evolutionary transitions in individuality”.

My point here is only that an appreciation of the history of the question and previous attempts to answer it should be given in the present paper. It is fine if the authors don't agree with my 1983 analysis, or find it limited, or different from theirs. Indeed, I find their analysis more satisfying. But some discussion should be given and they should clarify what the similarities and differences are in the two analyses with regard to answering the question of how the phenotype diverged from the genotype.

The author's say in their rebuttal. “Given that Michod's model is concerned with the origin of hypercycles, we feel that a comparison with his model is implicitly included in the new paragraphs mentioned above. If we missed something, we would be glad to know about it.”

To just say one (Michod 1983) is based on hypercycles and their analysis isn't doesn't go to the crux of the issue.

Here is what I think they have missed. There are two fitness components for a replicator, its capacity to encode information and its capacity to copy or replicate that information. The crux of the issue is simply that a replicator without an enzyme cannot simultaneously optimize both its information carrying capacity and the functional capacities that promote replication, when there are trade-offs involved between these two fitness components. By making an enzyme, the phenotype (functional capacities) can diverge from the genotype (information carrying capacity) and both can be optimized or enhanced independently. That is the essence of the model results (of course there is more) and the basic point I made in 1983.

We appreciate that the reviewer has asked many years ago in a general manner how the genotype and phenotype could diverge. Although our question is not identical because ours is concerned also with the unidirectionality of information flow from genomes to enzymes (the central dogma), the two questions certainly overlap. Thus, we added a discussion of the reviewer's work to our manuscript. However, we did this by comparing our work to the reviewer's 2006 PNAS paper. Below, we first describe the comparison. We then explain why we compare our work with the reviewer's 2006 paper rather than his 1983 paper.

To make our reply self-contained (for the editor), let us first recapitulate the reviewer's 2006 paper here. That paper describes a simple model that shows that reproductive division of labour, such as germ-soma differentiation, can evolve because it enables the independent optimization of an information-carrying capacity and a functional capacity. Importantly, the paper describes a necessary condition for this evolution to occur: a trade-off curve between an information-carrying capacity and a functional capacity must be convex. This condition is necessary because it generates a situation in which the separation of the two capacities maximises the fitness of a group (e.g., a group of cells constituting a multicellular organism). Only under this condition, does group-level selection favour the information-function separation. In short, the paper describes a type of trade-off relation for which reproductive division of labour is selectively beneficial.

In his comments, the reviewer suggests that symmetry breaking in our model is caused by the same mechanism as described by his 2006 PNAS paper. This is not the case. In the reviewer's model, reproductive division of labour is favoured by group-level selection. Contrariwise, in our model, reproductive division of labour is disfavoured by cellular-level selection because the fitness of a protocell is maximised if all replicators in the protocell are maximally catalytic. The trade-off relation in our model is such that replicators can optimise their catalytic activities without simultaneously compromising their templating ability if all replicators in a protocell provide an equal amount of catalysis. However, if replicators provide heterogenous amounts of catalysis, providing catalysis puts a replicator at a relative selective disadvantage within a protocell.

Consequently, cellular-level selection tends to maximise all catalytic activities of all replicators, whereas molecular-level selection tends to minimise all catalytic activities of all replicators. Therefore, for sufficiently high relatedness (i.e., for sufficiently small V and m), for which cellular-level selection is dominant, evolution produces replicators that have maximum catalytic activities to maximise the fitness of protocells—a state that involves no reproductive division of labour.

Another reason why our results are not explained by the mechanism suggested by the reviewer comes from the fact that symmetry breaking in our model occurs only for sufficiently low relatedness. This fact is completely opposite to the expectation from the theory of the reviewer's 2006 paper, which implies that reproductive division of labour evolves only if relatedness is high enough to ensure the effectiveness of group-level selection. We would also like to add that symmetry breaking in our model is not explained by Hamilton's rule as described in Page 12 Line 287-299 of our manuscript.

So, what selects reproductive division of labour in our model? The answer is “nothing.” Reproductive division of labour evolves, not because it is selected at any single level, but because it is a stable equilibrium between evolution driven by molecular-level selection and evolution driven by cellular-level evolution, an emergent outcome of conflicting multilevel selection, as revealed by our mathematical analysis.

We next explain why we think it is better to compare our work with the reviewer's 2006 PNAS paper instead of his 1983 paper. First, the 2006 paper describes a simple model that clearly and generally expresses the ideas outlined in the last paragraph of the reviewer's comments. We initially thought that the 2006 paper is specifically concerned with a germ-soma distinction because of the terminology used there. But, we have come to realise that the formulation of the model is general and that its scope actually includes a genome-enzyme distinction.

The second reason for comparing our work to the reviewer's 2006 paper is due to our failure to convince ourselves that what is described in the last paragraph of the reviewer's comments is actually the essence of the model results of the reviewer's 1983 paper. To make our reply self-contained again, let us briefly summarise the 1983 paper here. The

model described there is a standard kin-selection model, which is parametrised by three numbers: cost, benefit, and relatedness-like value (denoted by C , B , and F_{ST} , respectively). The model assumes two types of replicator: a hypercycle, which increases the fitness of replicators in the same local habitat by B at the fitness cost of C ; and a parasite, which does not benefit others, but receives benefit from the hypercycle. The model shows that the hypercycle can invade if $F_{ST}B > C$ —Hamilton's rule. The formulation of the model is sufficiently general for us to think its results are independent of whether or not there is a distinction between genomes and enzymes—i.e., whether the hypercycle encodes enzymes as conceived by the reviewer, or provides catalysis by itself as conceived in the RNA world hypothesis. Therefore, in our opinion, the model cannot inform us about whether or not a distinction between genomes and enzymes can evolve.

Our failure to convince ourselves is reinforced by the reviewer's 2006 PNAS paper. As mentioned above, that paper has shown that the convexity of a trade-off curve is necessary for the evolution of divergence between functional and information-carrying capacities through the mechanism outlined in the last paragraph of the reviewer's comments. This condition, however, is not imposed on the model of the reviewer's 1983 paper (i.e., no specific relation is assumed between B and C). Therefore, we think that it would be logically inconsistent if the 1983 model spoke about the divergence between functional and informational capacities: its results would have to imply that evolution can cause such divergence if $F_{ST}B > C$ irrespective of whether or not a trade-off curve is convex, a statement that contradicts the reviewer's 2006 paper. Such contradiction, however, does not arise if the model of the 1983 paper is more generally about the origin of a hypercycle, which does not necessarily imply separation between genotypes and phenotypes. In passing, we note that the reviewer's 1983 paper is not cited in his 2006 paper.

For the above reasons, we have compared our work specifically with the reviewer's 2006 paper, citing his 1983 paper as a historical reference (Page 13 Line 300-315). In addition, we added one sentence that elaborates on the trade-off relation in our model (Page 5 Line 109-111). We consider the comparison to the reviewer's work an important addition to our

manuscript because it exposes our theory's novelty from yet another angle. We express again our gratitude to the reviewer for helping us improve our manuscript.

Appendix C

Response to revision by Rick Michod.

The authors have still not addressed the second issue of my original review which was:

“The second issue concerns a paper by Michod referenced below which basically asks the same question as the present paper. This paper is extended in Chapters 2 and 3 of reference [5] in the present paper (see especially Fig 3-2 and related text). The present paper should clarify what are the similarities and differences in approach and results. R. E. Michod, Population biology of the first replicators: on the origin of the genotype, phenotype and organism. *Am. Zool.* 23, 5-14 (1983). “

In my second review I said what I think they have missed from the 1983 paper as it relates to their paper,

Here is what I think they have missed. There are two fitness components for a replicator, its capacity to encode information and its capacity to copy or replicate that information. The crux of the issue is simply that a replicator without an enzyme cannot simultaneously optimize both its information carrying capacity and the functional capacities that promote replication, when there are trade-offs involved between these two fitness components. By making an enzyme, the phenotype (functional capacities) can diverge from the genotype (information carrying capacity) and both can be optimized or enhanced independently. That is the essence of the model results (of course there is more) and the basic point I made in 1983.

In their rebuttal letter, the authors admit that my 1983 paper “certainly overlap[s]” with their present paper.¹ However, for some reason, they do not think the readers of their paper should know that. If I were writing a paper in which another worker had asked a question that most certainly overlapped with my question, I would want the readers to know that, I would want the reader to know what the other worker had done and how my paper related to theirs. It is still my opinion that the current submitted paper should do that.

The authors do not provide this context and comparison. Instead they now give the 1983 paper a gratuitous reference without substance (Page 13 Line 300-315) and briefly discuss a 2006 paper of mine about the evolution of multicellularity. I am not sure why the authors do not want to discuss the overlap with the 1983 paper. One reading of their rebuttal is that it is because they don't think I was asking the question I thought I was asking. For example, they say in their rebuttal that they reference the 2006 paper instead because this “is due to our failure to convince ourselves that what is described in the last paragraph of the reviewer's comments [the italicized remark above] is actually the essence of the model results of the reviewer's 1983 paper.”

There is a qualitative analysis in the 1983 paper that along with the model is the basis for the italicized remark above. Let me spell this out so there can be no mistaking the 1983 paper.

¹ They say in their rebuttal "We appreciate that the reviewer has asked many years ago in a general manner how the genotype and phenotype could diverge. Although our question is not identical because ours is concerned also with the unidirectionality of information flow from genomes to enzymes (the central dogma), the two questions certainly overlap."

Figure 1 of the 1983 paper and associated text on pp. 7-8 concerns how birth (replication) and death, the two basic life history components, may diverge when there is a replicator as opposed to spontaneous creation of short sequences. I go on to say that the replicator

“may serve as a template catalyst for the polymerization of a complementary strand or it may be degraded by hydrolysis into its mono-nucleotide components. The template reaction is the birth process of the replicator, so let b_i be the rate of polymerization of complementary strands per unit time. The hydrolysis of the molecule is the death process, so let d_i be the rate of this reaction per unit time.” (Michod 1983)

I go on to discuss and illustrate that without making proteins

“there are adaptive constraints on the birth and death process even at this primitive molecular level”. (Michod 1983)

It is these adaptive constraints or trade-offs that are key to my discussion that by making an enzyme the replicator may promote and optimize the birth and death process separately. For, without proteins, I say

“the phenotype and genotype are physically interrelated.” (Michod 1983)

Before constructing and analyzing the model for evolution of enzyme catalysis I conclude my qualitative analysis with the following summary.

“With the production of a protein, it was possible for the first time for the phenotype to diverge from the direct physical structure of the nucleotide sequence, or genotype. This, of course, was a major event in the evolution of life, allowing for the physical decoupling of function from the information which codes for the function. Previous to this event, function was inextricably tied to the genotype and its physicochemical properties. With their decoupling, increased flexibility would result in the promotion of replication and the protection of the replicator from hydrolytic decay. The remainder of the paper considers the selective pressures responsible for the production of proteins and the encapsulation of the genotype-protein complex in a proto-organism.” (Michod 1983)

The paper synopsis says

“At this prereplicator stage in the evolution of life there is no life history, since the birth and death processes are intimately coupled through the physical chemistry of a single reaction. With the emergence of nonenzymatic, template-directed replication, the birth and death processes could diverge for the first time, since selection could act differently on the birth and death rates of the replicating molecule. Thus, with replication, natural selection and life-history evolution began. The genotype, or nucleotide sequence, of the replicating molecule gave rise to several phenotypic properties, the most important of which was its three-dimensional structure which in turn affected the birth and death processes. However, at this stage of nonenzymatic template replication, the phenotype was the physical structure of the genotype, nothing more; For the divergence of the phenotype from the genotype it was necessary for the replicator to produce a protein.” (Michod 1983)

I could go on, but the above quotations should convince the reader that the 1983 paper supports the above italicized quote from my second review.

If the author's still do not wish to inform the reader of the overlap with 1983 paper, a paper which they admit "certainly overlap[s]" with their present paper, then I think the editor or another reviewer should arbitrate the issue as the authors and I will just go back-and-forth as we have three times now...

The 2006 paper of mine referred to in the rebuttal is primarily about the origin of multicellularity. It can be interpreted the way the authors do, but the 2006 paper was not especially about the origin of genotype and phenotype, while that was the whole point of the 1983 paper. This is why the 1983 paper wasn't referenced in the 2006 paper -- as the authors point out correctly in their rebuttal. Why the authors point this out, I don't understand.

Appendix D

(The comments of the editor and reviewer are italicised)

Associate Editor's Comments to the Authors:

As perhaps could be expected, the reviewer doesn't subscribe to your conclusion that his 1983 paper is about something different, and I agree with him. One cannot say that the current model is different from Michod's because it depends on conflicting levels of selection whereas Michod's 1983 analysis is based on kin selection. Maybe there is a subtle issue here, but in my mind kin and group selection were invented precisely to be able to study the consequences of conflicting levels of selection! I think the MS cannot be accepted without crediting Michod's 1983 study.

Author's Reply to Associate Editor's Comments:

The editor's comment made us realise that our previous rebuttal letter could be interpreted as conveying an unintended message. We did not intend to imply that the reviewer's 1983 paper is not about a genome-enzyme distinction because the analysis of his mathematical model is based on kin selection while ours is based on multilevel selection. Rather, we intended to say that the mathematical modelling described in the reviewer's 1983 paper is not about the evolution of a genome-enzyme distinction, but about the invasion of a hypercycle.

Reading the reviewer's comments below, we have come to gain a better understanding of the reviewer's 1983 work. We feel that we were misled about the actual contents of the reviewer's 1983 work as described in our rebuttal letter below. While we still maintain that the mathematical modelling described therein is not about the evolution of a genome-enzyme distinction, we also see another aspect of the reviewer's paper. With this understanding, we added a discussion of the reviewer's 1983 paper to our manuscript.

In this added discussion, we describe two major differences between our present work and the reviewer's (please see the revised manuscript for detail). Briefly,

1. our theory does not assume that a genome-enzyme distinction maximises the multiplication rates of replicators;
2. our model explicitly displays the evolution of a genome-enzyme distinction

through spontaneous symmetry breaking between replicators.

The first point above refers to the same difference as we pointed out in the discussion of the reviewer's 2006 paper. Therefore, the new discussion replaces the latter. Also, the editor's comment above led us to avoid referring to 'group-level selection' in order to minimise the chance of misleading the reader to think that our discussion is related to debates about whether group selection is the same as kin selection.

Response to revision by Rick Michod:

The authors have still not addressed the second issue of my original review which was:

“The second issue concerns a paper by Michod referenced below which basically asks the same question as the present paper. This paper is extended in Chapters 2 and 3 of reference [5] in the present paper (see especially Fig 3-2 and related text). The present paper should clarify what are the similarities and differences in approach and results. R. E. Michod, Population biology of the first replicators: on the origin of the genotype, phenotype and organism. Am. Zool. 23, 5-14 (1983).”

In my second review I said what I think they have missed from the 1983 paper as it relates to their paper,

Here is what I think they have missed. There are two fitness components for a replicator, its capacity to encode information and its capacity to copy or replicate that information. The crux of the issue is simply that a replicator without an enzyme cannot simultaneously optimize both its information carrying capacity and the functional capacities that promote replication, when there are trade-offs involved between these two fitness components. By making an enzyme, the phenotype (functional capacities) can diverge from the genotype (information carrying capacity) and both can be optimized or enhanced independently. That is the essence of the model results (of course there is more) and the basic point I made in 1983.

In their rebuttal letter, the authors admit that my 1983 paper “certainly overlap[s]” with their present paper. However, for some reason, they do not think the readers of their paper should know that. If I were writing a paper in which another worker had asked a question that most certainly overlapped with my question, I would want the

readers to know that, I would want the reader to know what the other worker had done and how my paper related to theirs. It is still my opinion that the current submitted paper should do that.

The authors do not provide this context and comparison. Instead they now give the 1983 paper a gratuitous reference without substance (Page 13 Line 300-315) and briefly discuss a 2006 paper of mine about the evolution of multicellularity. I am not sure why the authors do not want to discuss the overlap with the 1983 paper. One reading of their rebuttal is that it is because they don't think I was asking the question I thought I was asking. For example, they say in their rebuttal that they reference the 2006 paper instead because this "is due to our failure to convince ourselves that what is described in the last paragraph of the reviewer's comments [the italicized (quoted in this rebuttal letter) remark above] is actually the essence of the model results of the reviewer's 1983 paper."

There is a qualitative analysis in the 1983 paper that along with the model is the basis for the italicized (quoted in this rebuttal letter) remark above. Let me spell this out so there can be no mistaking the 1983 paper.

I They say in their rebuttal "We appreciate that the reviewer has asked many years ago in a general manner how the genotype and phenotype could diverge. Although our question is not identical because ours is concerned also with the unidirectionality of information flow from genomes to enzymes (the central dogma), the two questions certainly overlap."

Figure 1 of the 1983 paper and associated text on pp. 7-8 concerns how birth (replication) and death, the two basic life history components, may diverge when there is a replicator as opposed to spontaneous creation of short sequences. I go on to say that the replicator "may serve as a template catalyst for the polymerization of a complementary strand or it may be degraded by hydrolysis into its mono-nucleotide components. The template reaction is the birth process of the replicator, so let b_i be the rate of polymerization of complementary strands per unit time. The hydrolysis of the molecule is the death process, so let d_i be the rate of this reaction per unit time." (Michod 1983)

I go on to discuss and illustrate that without making proteins “there are adaptive constraints on the birth and death process even at this primitive molecular level”. (Michod 1983) It is these adaptive constraints or trade-offs that are key to my discussion that by making an enzyme the replicator may promote and optimize the birth and death process separately. For, without proteins, I say “the phenotype and genotype are physically interrelated.” (Michod 1983)

Before constructing and analyzing the model for evolution of enzyme catalysis I conclude my qualitative analysis with the following summary. “With the production of a protein, it was possible for the first time for the phenotype to diverge from the direct physical structure of the nucleotide sequence, or genotype. This, of course, was a major event in the evolution of life, allowing for the physical decoupling of function from the information which codes for the function. Previous to this event, function was inextricably tied to the genotype and its physicochemical properties. With their decoupling, increased flexibility would result in the promotion of replication and the protection of the replicator from hydrolytic decay. The remainder of the paper considers the selective pressures responsible for the production of proteins and the encapsulation of the genotype-protein complex in a proto-organism.” (Michod 1983)

The paper synopsis says “At this prereplicator stage in the evolution of life there is no life history, since the birth and death processes are intimately coupled through the physical chemistry of a single reaction. With the emergence of nonenzymatic, template-directed replication, the birth and death processes could diverge for the first time, since selection could act differently on the birth and death rates of the replicating molecule. Thus, with replication, natural selection and life-history evolution began. The genotype, or nucleotide sequence, of the replicating molecule gave rise to several phenotypic properties, the most important of which was its three-dimensional structure which in turn affected the birth and death processes. However, at this stage of nonenzymatic template replication, the phenotype was the physical structure of the genotype, nothing more; For the divergence of the phenotype from the genotype it was necessary for the replicator to produce a protein.” (Michod 1983)

I could go on, but the above quotations should convince the reader that the 1983 paper supports the above italicized quote from my second review. If the author’s still do not

wish to inform the reader of the overlap with 1983 paper, a paper which they admit “certainly overlap[s]” with their present paper, then I think the editor or another reviewer should arbitrate the issue as the authors and I will just go back-and-forth as we have three times now...

The 2006 paper of mine referred to in the rebuttal is primarily about the origin of multicellularity. It can be interpreted the way the authors do, but the 2006 paper was not especially about the origin of genotype and phenotype, while that was the whole point of the 1983 paper. This is why the 1983 paper wasn't referenced in the 2006 paper -- as the authors point out correctly in their rebuttal. Why the authors point this out, I don't understand.

Author's Reply to the Reviewer's Comments:

Reading the reviewer's comments, we have come to gain a better understanding of the actual contents of the reviewer's 1983 paper. While we still maintain that the mathematical modelling described therein is not about the evolution of a genome-enzyme distinction, we also see another aspect of the paper. With this understanding, we added a discussion of the reviewer's 1983 paper to our manuscript. Before describing this discussion, we would like to elaborate on why we did not discuss the reviewer's 1983 paper in our previous manuscript below.

In our past rebuttal letters, we always focused on the reviewer's mathematical modelling. This was in part because the reviewer specifically referred to his model results. For example, in his first-round comments, the reviewer directed us to pay special attention to Figure 3-2 of his 1999 book, which displays a result of his mathematical modelling. Also, in his second-round comments, the reviewer concludes, '*That is the essence of the model results (of course there is more) and the basic point I made in 1983.*'

As we have repeatedly stated in our past rebuttal letters, the mathematical modelling described in the reviewer's 1983 paper and its extension in his 1999 book are, by themselves, not about the evolution of a genome-enzyme distinction, but about the invasion of a hypercycle. In general, the invasion of a hypercycle does not necessarily imply the evolution of a genome-enzyme distinction. Moreover, there is no feature in

the reviewer's model that specifies that a hypercycle involves a genome-enzyme distinction—i.e., that a hypercycle is a non-catalytic template encoding an enzyme rather than a bifunctional replicator serving as both template and catalyst. Therefore, we have a grave doubt about the reviewer's claim that the model results of his 1983 paper are what is described in the reviewer's comment quoted below: *'There are two fitness components for a replicator, its capacity to encode information and its capacity to copy or replicate that information. The crux of the issue is simply that a replicator without an enzyme cannot simultaneously optimize both its information carrying capacity and the functional capacities that promote replication, when there are trade-offs involved between these two fitness components. By making an enzyme, the phenotype (functional capacities) can diverge from the genotype (information carrying capacity) and both can be optimized or enhanced independently. That is the essence of the model results (of course there is more) and the basic point I made in 1983.'* We did not discuss the reviewer's 1983 paper in our previous manuscript because the mathematical modelling described there is not about the evolution of a genome-enzyme distinction, as we stated in our previous rebuttal letter.

In his third-round comments above, the reviewer refers to 'qualitative analysis'. The reviewer writes, *'There is a qualitative analysis in the 1983 paper that along with the model is the basis for the italicized remark above,'* and *'Before constructing and analyzing the model for evolution of enzyme catalysis I conclude my qualitative analysis with the following summary.'* From these comments, we gathered that 'qualitative analysis' refers to the verbal arguments in the reviewer's 1983 paper that precede and, therefore, are not conclusions drawn from his mathematical modelling. This realisation led us to shift our focus and better understand the actual logical structure of the reviewer's 1983 paper.

We have come to recognise that the reviewer's 1983 paper includes at least two distinct parts: verbal arguments for the benefit of a genome-enzyme distinction, and model results about the invasion of a hypercycle. More specifically, the reviewer's 1983 paper is structured as follows:

1. The reviewer hypothesised that a genome-enzyme distinction maximises the multiplication rates of replicators because it allows the unconstrained

optimisation of the replication rate and hydrolytic resistance that are in a trade-off relation.

2. Assuming that the above hypothesis is true, how can a system involving a genome-enzyme distinction be selected for? To address this question, the reviewer used mathematical modelling to examine the possibility of the invasion of a hypercycle, assuming that a hypercycle involves a genome-enzyme distinction, an assumption that is external to the model. His model results show that a hypercycle can invade if a version of Hamilton's rule is satisfied.

With this understanding, we recognise that there are two important differences between the reviewer's 1983 work and our present work, each relating to one of the above points, as briefly summarised below (please see our revised manuscript for detail). First, our theory does not assume that a genome-enzyme distinction maximises the multiplication rates of replicators. Second, our model explicitly displays the evolution of a genome-enzyme distinction through spontaneous symmetry breaking between replicators.

One might wonder how our model could display the evolution of a genome-enzyme distinction without the assumption that a genome-enzyme distinction maximises the multiplication rates of replicators. The answer is a positive feedback between reproductive values and the relative impact of selection at different levels, as demonstrated mathematically and numerically in our manuscript. Moreover, the evolution of a genome-enzyme distinction occurs only if relatedness is sufficiently low, a result that is diametrically opposite to the reviewer's result under the assumption that a hypercycle involves a genome-enzyme distinction.

In the revised manuscript, we have added a discussion of the reviewer's 1983 paper describing the above two differences. The difference about the nature of the trade-off constitutes essentially the same point as we made in the discussion of the reviewer's 2006 paper in our previous manuscript. Therefore, the former replaces the latter.

Appendix E

(The comments of the associate editor are italicised)

Associate Editor's comments

This MS has gone through a few rounds already but seems to be stuck in a standoff between the authors and a reviewer, Rick Michod. I was quite reluctant to send it yet again to the reviewer and decided to see whether I fully understood what the standoff is about. Michod claims that this MS addresses the same question as one of his early (1983) paper; the authors feel this paper deals with something different. They did cite a later paper by Michod which is indeed relevant but this does not address the same question (the reference has gone from the present version). Looking up Michod (1983), I have to conclude that it indeed addresses a very similar question as the present MS:

"The purpose of this article is to study the selective pressures responsible for the divergence of the phenotype from the genotype, and the origin of the cellular organism."

The way the authors deal with it is indeed rather different and, as also Michod concedes, goes to a more fundamental level. As the authors note in their rebuttal letter, Michod (1983) studies various hypercycle models which takes the existence of separate catalysers for granted whereas the present MS allows these to emerge by symmetry breaking. I consider these results novel and worthy of publication in the Proceedings. However, I still have trouble with the way they, reluctantly it seems, deal with Michod's early paper.

In the preceding back-and-forth the authors claimed the main difference between the two approaches was that Michod's results depend on kin selection whereas their finding results from "conflicting multilevel selection". I am aware that there is some controversy regarding these mechanisms but for the present issue this is irrelevant: though Michod discusses his results in kin selection terms, the model he analyses is a proper group selection (= multilevel) model.

In the last revision the authors avoided the kin selection-multilevel selection issue, which is very welcome, and added a discussion of Michod's work which is also nice. Although I do not fully agree with all they claim, to me this is not really a problem, considering it is up to them to defend their views if it stirs up a polemic. However, Michod's paper is only the 36th to be cited, toward the very end of the discussion, and I think it should be mentioned much earlier.

Although I do like the MS and the analysis it presents and I do think it will have a significant impact if published, I still hesitate to recommend it. If the authors acknowledge in the introduction that the question has been addressed before but that their approach delivers some novel insights all would be fine with me.

Author's replies

Following the suggestion of Associate Editor, we cited Michod's 1983 paper in Introduction as the 3rd reference. There, we state, "Michod hypothesised that a genome-enzyme distinction evolved because the distinction maximised the multiplication rates of replicators by allowing the unconstrained optimisation of the replication rate and hydrolytic resistance of replicators that are in a trade-off relation." After this statement, we say that we explore an alternative possibility.

We agree with Associate Editor that the controversy regarding kin selection and multilevel selection is irrelevant to our discussion of Michod's work, as we stated in our previous reply. In addition, we would like to mention one of the plainest ways to see the difference between Michod's results and ours might, as follows. Michod's model shows that a hypercycle invades if relatedness is sufficiently **high** (as stated by Associate Editor, we also think that Michod took it for granted that the hypercycle involves a genome-enzyme distinction). By contrast, our model shows that a genome-enzyme distinction evolves if relatedness is sufficiently **low**—a result that is diametrically opposite to that of Michod's model. The same difference is explicitly stated in the part of Discussion where we compare kin selection theory and our theory. This statement is referred to at the end of our discussion of Michod's work.